# Dueling in the Dark: An Efficient and Optimal Mirror Descent Approach for Online Convex Optimization with Adversarial Preferences

## Abstract

Recent developments in Large Language Models (LLMs) have sparked significant attention in Reinforcement Learning from Human Feedback (RLHF). A simple, widely used, and cost-effective method for gathering human feedback is through relative queries based on human preferences, where the pairwise preference of two alternatives is often modeled as the sigmoid of their respective utility scores. Despite the popularity of these sigmoid-based RLHF frameworks, their theoretical foundations remain underdeveloped as existing algorithms often lack the desired performance guarantees, or are limited to small-scale problems due to computationally intractable steps. We address this challenge by developing the first efficient online gradient descent-based algorithm for the problem with provably optimal performance guarantees. In fact, our proposed methods work even for adversarially changing preferences, unlike existing attempts, which assume a fixed underlying stochastic preference model. Formally, we consider the adversarial online convex (linear) optimization (OLO) problem in $d$-dimensions, but unlike the existing OLO framework, we assume only that the learner can observe a (weaker) preference feedback upon choosing a few alternatives at each round. With the objective of identifying the best arm, we propose an efficient online mirror descent (OMD) based approach for the problem with regret and sample complexity guarantees. The main challenge lies in finding a suitable gradient approximation of the underlying (adversarially changing) utility functions solely from the weak preference feedback, as opposed to the conventional gradient or value feedback used in OLO. We also extend our methods beyond pairwise preferences to multi-way preference ($B$-sized batched pairwise) and partial ranking feedback with improved performance guarantees. Additionally, our algorithms are optimal as we proved by matching lower bounds closing the potential of any better algorithms for the settings. Our contribution lays the groundwork for a practical gradient descent-based algorithm in RLHF. Supported by robust theoretical guarantees, our approach holds promise in the current landscape of developing efficient algorithms for LLMs and addressing human-AI alignment challenges.

## 1 Introduction

The rapidly advancing field of AI has sparked interest in Reinforcement Learning from Human Feedback (RLHF), which incorporates human input to refine AI systems, mitigating risks in autonomous decision-making and fostering systems that act aligned with users' best interests. This paper explores the theoretical aspects of RLHF with preference feedback, emphasizing its potential to enhance AI alignment.

Human preference feedback is a critical form of feedback within the field of machine learning (ML). Unlike conventional feedback models used in ML optimization literature for designing predictive AI models, which includes demonstration Hussein et al. (2017); Swamy et al. (2023); Torabi et al. (2018, gradient-based Zinkevich (2003); Boyd et al. (2004); Fletcher (2013), value-based feedback Flaxman et al. (2005); Cesa-Bianchi & Lugosi (2006b); Shamir (2015); Saha (2021a), preference feedback is a *weaker form of feedback* that receives only relative desirability (a.k.a. preference) of different outcomes/actions for a given task. However, on the positive end, preference feedback can

capture a more nuanced understanding of human values and priorities by tallying the relative desirability of different outcomes. Studies in psychology and cognitive neuroscience also corroborate the fact that humans are often naturally more comfortable providing relative feedback compared to the other modes (Musallam et al., 2004; Kahneman & Tversky, 1982), hence the training data tend to be less biased and resource-efficient. Consequently, this form of feedback enables AI systems to learn more complex and subtle aspects of human intentions, which are often difficult to encode through demonstrations or reward feedback. This also makes RLHF with preference feedback a powerful tool for improving the reliability and safety of AI systems in practice.

To understand the RLHF with preference feedback problem Wu & Sun (2024); Xie et al. (2024); Xiong et al. (2024); Rafailov et al. (2024) more formally: In the simple online/active exploration RLHF with preference feedback setting, the learner can sequentially query a pair of actions and receive binary 0–1 preference feedback indicating the preferred item. The objective for these classes of problems is usually to find a good (value-maximizing) policy $\pi : \mathcal{C} \mapsto \mathcal{D}$, a mapping from the context space $\mathcal{C}$ to decision space $\mathcal{D}$, as efficiently as possible. The decision space $\mathcal{D}$ represents the set of actions/alternatives to learn from, e.g., for language models $\mathcal{D}$ could be the class of all words (or tokens), the set of trajectories for autonomous car driving, or the set of movies for a movie-recommender system, etc.

Existing work on preference-based learning for online (exploratory) RLHF, whether empirical or theoretical, is limited by computationally inefficient algorithms (Xie et al., 2024; Xiong et al., 2024). Many current approaches struggle to scale effectively with the complexity of real-world scenarios, often requiring extensive computational resources and time to process human feedback and update AI models accordingly. This inefficiency not only hampers the practical deployment of preference-based learning systems but also restricts their ability to quickly adapt to dynamic environments and evolving human preferences.

**Limitations of Existing Online RLHF with Preference Feedback Algorithms.** The well-cited work of Rafailov et al. (2024); Ouyang et al. (2022); Chen et al. (2024) use offline data in nature which does not allow active exploration and also lack convergence guarantees. Recently the literature saw a surge of papers on online RLHF with preference feedback (Xu et al., 2020; Chatterji et al., 2021; Saha et al., 2023; Saha, 2021a; Kausik et al., 2024; Das et al., 2024), however, these algorithms are based on the optimism in the face of the uncertainty (UCB based) principle which requires maintaining confidence sets and optimizing over the policy space which could be computationally intractable. Few studies (Efroni et al., 2021; Li et al., 2024; Wu & Sun, 2024) have also considered Thompson Sampling (TS) approaches as an alternative but again updating and sampling from the posterior distribution could be computationally hard as well, making them impractical for real-world applications. Quoting from Xie et al. (2024), "However, the most powerful approaches in this space are computationally intractable in the general reinforcement learning (RL) setting (Jiang et al., 2017; Jin et al., 2021; Foster et al., 2021), and prior attempts to adapt them to RLHF either make unrealistic modeling assumptions (i.e., do not allow for general function approximation) (Xu et al., 2020; Novoseller et al., 2020; Saha et al., 2023; Wu & Sun, 2024; Zhan et al., 2023; Du et al., 2024; Das et al., 2024), or are computationally inefficient and not feasible to faithfully implement (Chen et al., 2022; Wang et al., 2023; Ye et al., 2024)," which nicely summarizes the state of the literature. In fact, the computational efficiency of Xie et al. (2024) itself is in question since they require to optimize in the policy space using methods like PPO (Schulman et al., 2017) which might not be runtime efficient unless the policy space is finite or parameterized under some restrictive assumptions.

Consequently, there is a pressing need for the development of more computationally efficient algorithms that can harness preference feedback in a timely and resource-effective manner, thereby enhancing the feasibility and responsiveness of AI alignment strategies. In this work, we present the first mirror (gradient) descent-based algorithm for the problem with an optimal performance guarantee.

## 1.1 CONTRIBUTIONS

For simplicity, we frame the RLHF problem as a best-arm identification problem in the online linear optimization (OLO) framework Shamir (2015); Hazan (2019) in dimension $d$, with pairwise preference feedback (see Sec. 2). Note this is the first step in designing gradient descent-based approaches

for RLHF which certainly will play a critical role in extending these methods for more complex policy optimization settings.

**Key Contribution.** Our contributions are multifaceted. At a technical level, we are the first to address the problem of adversarial online linear optimization with preference feedback, which has critical implications in the RLHF with preference feedback literature. Our specific contributions are: •(1) **Algorithmic Contribution.** Our key contribution is to design an online mirror-descent based (OMD) algorithm to obtain a near-optimal $\tilde{O}(d\sqrt{T})$ regret algorithm for online optimization with adversarial preferences (Alg. 1,Thm. 1). Our algorithm is motivated by the *Scrible* algorithm of Hazan (2019). However, *Scrible* operates under 'value feedback', as opposed to preference feedback of our setting, which allowed them to use the standard 'one-point gradient estimation' technique (Flaxman et al., 2005) to estimate the loss functions. One of our primary contributions in our proposed algorithm *Double-Scrible* is to estimate the loss function per round from the *weaker preference feedback*. Consequently, we had to adapt to a different proof analysis to incorporate the changes which finally led to near-optimal (upto log factors) $\tilde{O}(d\sqrt{T})$ regret bounds of *Double-Scrible* (Thm. 1). •(2) **Performance Limit Analysis.** To understand the effectiveness of our analysis, we further prove a matching lower bound of $\Omega(d\sqrt{T})$ to show that our algorithm is within a logarithmic factor of the optimal performance limit (Thm. 3). Deriving the lower bound for this problem was non-trivial, we derive this from the first principle lifting tools from the classical literature of information theory.

**Additional Contributions.** We enriched and extended our above result with pairwise preference in multiple ways:

1. In Sec. 4, we first generalize the above algorithm to multiwise (batched) preference feedback, where the learner can query a set of $B$ pairwise preferences in one go. In many settings, it is not feasible to actively update the model's prediction after every round, perhaps due to communication delays, parallel processing or time/ cost overhead. Instead, the system may prefer to collect a bunch of comparison queries in a batch and then update its model to generate the next set of queries. In such settings, batched RLHF is a natural model to consider. Our improved analysis of Alg. 2, the batched variant of *Double-Scrible* shows that one can achieve a faster $\tilde{O}(\frac{d}{\sqrt{\min\{B,d\}}}\sqrt{T})$ regret learning rate for this case (Thm. 4).

2. Next we consider another interesting feedback model in Sec. 5 which generalizes pairwise preference feedback to subsets of size $k$ (for any $k \geq 2$), and allows the learners to query partial rank ordered feedback of length $m \in \{1, \ldots k\}$. The objective was to understand if the learning algorithm is allowed to query from a larger set of $k$ alternatives and obtain a richer $m$-length ranking feedback, can it learn faster? What is the optimal trade-off of the learning rate with $m$ and $k$? Our proposed algorithm *MNL-Scrible* addresses this setting with a regret guarantee of $\tilde{O}(\frac{d}{\sqrt{\min\{m,d\}}}\sqrt{T})$. The $k$ (subsetsize) independence of the result could be surprising to many as one may expect that larger subsets may lead to faster convergence! However, this is not the case we explained in Rem. 6. On the other hand, as expected the regret indeed improves with the increasing length of the rank-ordered feedback $m$. Our algorithm *MNL-Scrible* actually exploits the key ideas of our batched algorithm Alg. 2 by *cleverly extracting $m$-batched pairwise preference information from the Top-$m$ ranking feedback $\sigma_{m,t}$*. We describe the algorithm in Sec. 5 and its regret performance follows almost immediately from Thm. 4.

3. Same as Thm. 3, we corroborate the performance analysis of all our algorithms with their corresponding lower bound analysis to understand the tightness of our algorithmic guarantees. Precisely, for the $B$-batched feedback setting we show that our regret guarantee of *BaBle-Scrible* is optimal (up to log factors) with a matching lower bound analysis Thm. 6. Similarly, our Top-$m$ ranking feedback lower bound of Thm. 9 justifies the tightness of regret performance of *MNL-Scrible*. All our lower bounds are derived from the first principles of information theory.

**Advantage of Gradient Descent Methods:** Gradient-based methods have multiple advantages compared to confidence-based methods: (1) GD/OMD handle high-dimensional problems efficiently due to their reliance on gradient information: (2) They are suitable for both stochastic and adversarial environments, making the gradient-based methods robust to changing data distributions or the underlying loss/reward functions which is often more practical for modeling real-world prob-

lems, (3) These methods can optimize a wide range of objective functions, including non-linear, non-convex, and constrained problems, (4) Gradient descent algorithms are simple to implement, even seamlessly integrate with modern deep learning frameworks, making these methods computationally efficient, unlike many UCB and TS based methods which often do not have a closed form solution Saha et al. (2023); Das et al. (2024) or sampling from the posteriors could be complicated Novoseller et al. (2020), and (5) Gradient descent techniques are inherently robust to model misspecification and smoothly integrate with differential privacy techniques.

## 2    PROBLEM SETUP

**Notation.**   Let $[n] = \{1, \ldots n\}$, for any $n \in \mathbb{N}$. Given a set $S$ and two items $x, y \in S$, we denote by $x \succ y$ the event $x$ is preferred over $y$. For any $r > 0$, let $\mathcal{B}_d(r)$ and $\mathcal{S}_d(r)$ denote the ball and the surface of the sphere of radius $r$ in $d$ dimensions respectively. $\mathbf{I}_d$ denotes the $d \times d$ identity matrix. For any vector $\mathbf{x} \in \mathbb{R}^d$, $\|\mathbf{x}\|_2$ denotes the $\ell_2$ norm of vector $\mathbf{x}$. $\mathbf{1}(\varphi)$ is generically used to denote an indicator variable that takes the value 1 if the predicate $\varphi$ is true and 0 otherwise. $\text{Unif}(S)$ denotes a uniform distribution over any set $S$. We write $\tilde{O}$ for the big O notation up to logarithmic factors. For any set $\Omega \subset \mathbb{R}^d$, $\text{int}(\Omega)$ denotes the interior of the set $\Omega$. $\text{Ber}(p)$ defines *Bernoulli* distribution with parameter $p \in [0, 1]$.

### 2.1    PROBLEM: ADVERSARIAL LOGISTIC DUELING BANDITS (LOGIT-DB):

We consider an online $T$ round sequential decision-making game on a decision space $\mathcal{D} \subset \mathbb{R}^d$ in the *Adversarial Online Linear Optimization (Bandits)* Hazan (2019); Abernethy et al. (2008) framework. At every round, the algorithm plays $\mathbf{x}_t, \mathbf{y}_t \in \mathcal{D}$ and observes a binary feedback $o_t$ s.t.

$$o_t \sim \text{Ber}\Big(\sigma\big(\boldsymbol{\theta}_t^{*\top}(\mathbf{x}_t - \mathbf{y}_t)\big)\Big).$$

We denote the probability of arm $\mathbf{x}$ being preferred over arm $\mathbf{y}$ as:

$$P_t(\mathbf{x}, \mathbf{y}) = \sigma\big(\boldsymbol{\theta}_t^{*\top}(\mathbf{x} - \mathbf{y})\big) = \frac{\exp(\boldsymbol{\theta}_t^{*\top}\mathbf{x})}{\exp(\boldsymbol{\theta}_t^{*\top}\mathbf{x}) + \exp(\boldsymbol{\theta}_t^{*\top}\mathbf{y})}, \quad \forall \mathbf{x}, \mathbf{y} \in \mathcal{D}.$$

Note we call the problem LOGIT-DB since the preference relation $P_t$ follows a logistic model, as $\sigma : \mathbb{R} \mapsto [0, 1]$ is the logistic link function, i.e. $\sigma(x) = (1 + e^{-x})^{-1}$.

**Objective-I: Regret Minimization w.r.t. the Best Choice.**   The goal of the algorithm is to minimize the cumulative regret, defined as:

$$\text{Reg}_T^{\text{Logit-DB}} := \sum_{t=1}^{T} \left[ \frac{(P_t(\mathbf{x}^*, \mathbf{x}_t) - 1/2) + (P_t(\mathbf{x}^*, \mathbf{y}_t) - 1/2)}{2} \right],$$

assuming $\mathbf{x}^* \leftarrow \arg\max_{\mathbf{x} \in \mathcal{D}} \sum_{t=1}^{T} \boldsymbol{\theta}_t^{*\top} \mathbf{x}$ the best (highest scoring) arm in the hindsight.

.

**Remark 1.** *For any $x \in \mathcal{D}$, note then $\frac{\boldsymbol{\theta}_t^{*\top}(\mathbf{x}^* - \mathbf{x})}{4} \leq P_t(\mathbf{x}^*, \mathbf{x}) - 1/2 \leq \boldsymbol{\theta}_t^{*\top}(\mathbf{x}^* - \mathbf{x})$, when $\mathcal{D} \subseteq \mathcal{B}_d(1)$. We prove this in App. A. Consequently, in the rest of the paper, we will address the regret*

$$\widehat{\text{Reg}}_T^{\,Logit\text{-}DB} := \sum_{t=1}^{T} \left[ \frac{\boldsymbol{\theta}_t^{*\top}(\mathbf{x}^* - \mathbf{x}_t) + \boldsymbol{\theta}_t^{*\top}(\mathbf{x}^* - \mathbf{y}_t)}{2} \right],$$

*noting $\text{Reg}_T^{Logit\text{-}DB} \leq \widehat{\text{Reg}}_T^{\,Logit\text{-}DB}$ from Rem. 1, thus designing algorithm to bound $\widehat{\text{Reg}}_T^{\,Logit\text{-}DB}$ would suffice to bound $\text{Reg}_T^{Logit\text{-}DB}$.*

**Objective-II: Sample Complexity.**   One can also consider a different learning objective where instead of regret minimization the goal is to find '$\epsilon$-best arm'. Precisely, we define an arm $\hat{\mathbf{x}} \in \mathcal{D}$ to be an *$\epsilon$-best arm* if $\frac{1}{T} \sum_{t=1}^{T} \boldsymbol{\theta}_t^{*\top}(\mathbf{x}^* - \hat{\mathbf{x}}) < \epsilon$. The goal could be find any such $\hat{\mathbf{x}} \in \mathcal{D}$ with the least number of samples.

**Remark 2.** *An attentive reader might have already noticed that for our problem setting,* regret minimization *is a stronger objective than the latter as a regret guarantee of a learning algorithm immediately yields a valid* sample complexity *bound for the same setting* $\hat{\mathbf{x}} = \frac{1}{T}\sum_{t=1}^{T}\frac{(\mathbf{x}_t + \mathbf{y}_t)}{2}$.

To understand Rem. 2 more formally, consider any algorithm $\mathcal{A}$ with regret bound $\widehat{\text{Reg}}_T^{\text{Logit-DB}}(\mathcal{A}) \leq R_{\mathcal{A}}(T)$. Then

$$
\frac{1}{T}\sum_{t=1}^{T} \boldsymbol{\theta}_t^{*\top}(\mathbf{x}^* - \hat{\mathbf{x}}) = \frac{1}{T}\sum_{t=1}^{T} \boldsymbol{\theta}_t^{*\top}\left[\mathbf{x}^* - \frac{1}{T}\sum_{t=1}^{T}\frac{(\mathbf{x}_t + \mathbf{y}_t)}{2}\right]
$$

$$
= \frac{1}{T^2}\sum_{t=1}^{T}\left[T\boldsymbol{\theta}_t^{*\top}\mathbf{x}^* - \sum_{t=1}^{T}\frac{(\mathbf{x}_t + \mathbf{y}_t)}{2}\right] = \frac{1}{T}\sum_{t=1}^{T}\left[\boldsymbol{\theta}_t^{*\top}\mathbf{x}^* - \sum_{t=1}^{T}\frac{(\mathbf{x}_t + \mathbf{y}_t)}{2}\right] \leq \frac{R_{\mathcal{A}}(T)}{T}.
$$

Let use define $g(a) = R_{\mathcal{A}}(a)/a$ for any $a \in \mathbb{N}$, then equating $g(T) \leq \epsilon$, the desired sample complexity $T = g^{-1}(\epsilon)$.

## 2.2 PROBLEM SETUP: ADVERSARIAL BATCHED LOGIT-DB

As motivated in the introduction, from a practical viewpoint it might be hard to actively update the parameters of the learning algorithm at each round $t$ due to time or cost overhead. Distributed deployment of the system might also hinder an active sequential adaptation of the learning algorithm due to parallel processing. Thus a natural variant of Logit-DB problem is to consider a *batched feedback model* where the learner gets to (actively) query $B$-pairwise queries in a batched fashion: Precisely, at each round $t$, the learner gets to query $B$-pairs $\{(\mathbf{x}_t^1, \mathbf{y}_t^1), (\mathbf{x}_t^2, \mathbf{y}_t^2), \ldots, (\mathbf{x}_t^B, \mathbf{y}_t^B)\}$ together and observes the corresponding $B$-pairwise preferences $o_t^1, o_t^2, \ldots o_t^B$, where $o_t^i \sim \text{Ber}\big(\sigma\big(\boldsymbol{\theta}_t^{*\top}(\mathbf{x}_t^i - \mathbf{y}_t^i)\big)\big)$.

**Regret Objective.** Same as $\widehat{\text{Reg}}_T^{\text{Batched-LogitDB}}$, the objective of the learner, in this case, is to minimize the regret over $T$ rounds, defined as:

$$
\widehat{\text{Reg}}_T^{\text{Batched-LogitDB}} := \sum_{t=1}^{T}\frac{1}{B}\left[\sum_{i=1}^{B}\frac{\boldsymbol{\theta}_t^{*\top}(\mathbf{x}_t^* - \mathbf{x}_t^i) + \boldsymbol{\theta}_t^{*\top}(\mathbf{x}_t^* - \mathbf{y}_t^i)}{2}\right].
$$

Note we can also consider the exact same *Sample complexity* objective in this setting as well, as defined above for Logit-DB in Sec. 2.1.

## 2.3 PROBLEM SETUP: ADVERSARIAL MULTINOMIAL LOGIT BANDITS (MNL-BANDITS):

In this setting, we generalize the Logit-DB problem (Sec. 3) to the subsetwise case, where the learner can observe preference feedback over a subset of items. More formally, as before, we consider a decision space $\mathcal{D} \subset \mathbb{R}^d$ and at every round the algorithm plays a $k$-subset $S_t := \{\mathbf{x}_t^1, \mathbf{x}_t^2, \ldots \mathbf{x}_t^k\} \subseteq \mathcal{D}$ and gets to see *Categorial* feedback $o_t \in [k]$ indicating the index of the winning arm in $S_t$ s.t. $o_t \sim \text{Categorical}\big(p_1^t, p_2^t, \ldots, p_k^t\big)$, where $p_i^t = \frac{\exp(\boldsymbol{\theta}_t^{*\top}\mathbf{x}_t^i)}{\sum_{j=1}^{k}\exp(\boldsymbol{\theta}_t^{*\top}\mathbf{x}_t^j)}$.

In fact, one could even consider a top-$m$ ranking generalization of the above feedback model. Let for any subset $S$, $\Sigma_S = \{\sigma \mid \sigma \text{ is a permutation over items of } S\}$, where for any permutation $\sigma \in \Sigma_S$, $\sigma(i)$ denotes the element at the $i$-th position in $\sigma$, $i \in [|S|]$. Then one can further define $\Sigma_S^m := \{\sigma_m := (\sigma(1), \ldots, \sigma(m)) \mid \sigma \in \Sigma_S\}$ and formally define the top-$m$ ranking feedback as follows:

**Generalized Top-$m$ ranking of items (TR-$m$):** In this case at every round $t$ the environment returns a ranking of the top $m$ items from $S_t$ ($|S_t| = k$) by drawing a full ranking $\sigma_t \in \Sigma_{S_t}$ over $S_t$ according to Plackett-Luce (PL) model without replacement, and returns the first $m$ ranked elements of $\sigma$, i.e., $(\sigma(1), \ldots, \sigma(m))$. More precisely, if $\sigma_{m,t}$ is any random top-$m$ ranking on a $k$-subset $S_t$ drawn according to the multinomial (MNL) model (Saha & Gopalan, 2019), then for every position $i \in [m]$, $1 \leq m \leq k - 1$,

$$
\sigma_{m,t}(i) \sim \text{Categorical}\big(p_1^t, p_2^t, \ldots, p_k^t \setminus \{p_{\sigma_{m,t}(1)}^t, p_{\sigma_{m,t}(2)}^t, \ldots p_{\sigma_{m,t}(i-1)}^t\}\big), \text{ s.t. } p_i^t = \frac{\exp(\boldsymbol{\theta}_t^{*\top}\mathbf{x}_t^i)}{\sum_{j=1}^{k}\exp(\boldsymbol{\theta}_t^{*\top}\mathbf{x}_t^j)},
$$

is drawn by successively sampling $m$ winners from $S$ according to the PL model, without replacement. Thus (equivalently) we have:

$$\mathbf{P}_t(\sigma_{m,t}|S) = \prod_{i=1}^{m} \frac{\exp(\boldsymbol{\theta}_t^{*\top}\mathbf{x}_t^{\sigma_{m,t}(i)})}{\sum_{j=i}^{m}\exp(\boldsymbol{\theta}_t^{*\top}\mathbf{x}_t^{\sigma_{m,t}(j)}) + \sum_{j\in S_{t\setminus m}}\exp(\boldsymbol{\theta}_t^{*\top}\mathbf{x}_t^{\sigma_{m,t}(j)})},$$

$$\text{such that } S_{t\setminus m} = S_t \setminus \{\sigma_{m,t}(i)\}_{i=1}^{m}.$$

Note for $k = 2$, the top-$m$ ranking feedback is equivalent to the dueling feedback discussed in Sec. 2.1.

**Regret Objective.** Following the objective from `Logit-DB` from Sec. 2.1 and Rem. 1, one can similarly define the regret for this setup as:

$$\widehat{\text{Reg}}_T^{\text{MNL}} := \sum_{t=1}^{T}\left(\frac{1}{k}\sum_{\mathbf{x}\in S_t}\boldsymbol{\theta}_t^{*\top}(\mathbf{x}^* - \mathbf{x})\right),$$

again assuming $\mathbf{x}^* := \arg\max_{\mathbf{x}\in\mathcal{D}}\sum_{t=1}^{T}\boldsymbol{\theta}_t^{*\top}\mathbf{x}$ the best arm in the hindsight.

## 3 DUELING CASE: ALGORITHM FOR LOGIT-DB PROBLEM

In this section, we investigate the `Logit-DB` problem (Sec. 2.1) for the pairwise preference (dueling) feedback.

**Algorithm description:** Our algorithm is motivated by the Scrible algorithm from Abernethy et al. (2008); Hazan (2019), which is a variant of the online mirror descent algorithm with a self-concordant barrier Boyd et al. (2004) as the regularizer.[1] The algorithm iteratively updates the decision variable $\mathbf{w}_t$ by minimizing the sum of the $\psi$-regularized linearized loss within the $\delta$-contracted decision set $\mathcal{D}_\delta := \{\mathbf{x} \mid \frac{1}{1-\delta}\mathbf{x} \in \mathcal{D}\}$. Precisely at each step $t$, we compute $\mathbf{w}_t = \arg\min_{\mathbf{w}\in\mathcal{D}_\delta}\left\{\eta\sum_{\tau=1}^{t-1}\mathbf{g}_\tau^\top\mathbf{w} + \psi(\mathbf{w})\right\}$. We then perform eigendecomposition of the Hessian $\nabla^2\psi(\mathbf{w}_t)$, sample an index $i_t$ uniformly at random from $[d]$, and generate perturbed solutions $\mathbf{x}_t = \mathbf{w}_t + \gamma_t\frac{1}{\sqrt{\lambda_{t,i_t}}}\mathbf{v}_{t,i_t}$ and $\mathbf{y}_t = \mathbf{w}_t - \gamma_t\frac{1}{\sqrt{\lambda_{t,i_t}}}\mathbf{v}_{t,i_t}$. It is important here to note that $\mathbf{x}_t, \mathbf{y}_t \in \mathcal{D}$ owing to the properties of self-concordant barrier functions, as argued in Lem. 10. By playing the pair $(\mathbf{x}_t, \mathbf{y}_t)$ and observing the outcome $o_t$, we construct the gradient estimator $\mathbf{g}_t = \frac{d}{\gamma_t}(o_t - \frac{1}{2})\sqrt{\lambda_{t,i_t}}\mathbf{v}_{t,i_t}$ for the next iteration and continue to the step iteration. Thm. 1 analyze the regret performance of Alg. 1 yielding an optimal $O(\sqrt{T})$ regret for the problem, as justified in Rem. 4. Due to space limitations, the algorithm pseudocode is given in App. B.1 and the detailed regret analysis of *Double-Scrible* (Alg. 1) is given in App. B.3.

**Theorem 1** (Regret Analysis of Alg. 1). *Consider the decision space $\mathcal{D}$, such that $\nabla^2\psi(\mathbf{w}) \geq H_{\mathcal{D},\psi}^2\mathbf{I}_d, \forall\mathbf{w} \in \mathcal{D}$. Then for the choice of $\eta = \frac{\sqrt{\nu}H_{\mathcal{D},\psi}}{d\sqrt{T\log T}}$, $\delta = \frac{1}{T}$ and $\gamma_t \leq 0.7H_{\mathcal{D},\psi}$, the Double-Scrible (Alg. 1) guarantees a regret bound:*

$$\widehat{\text{Reg}}_T^{\text{Logit-DB}} \leq O\left(\frac{d\sqrt{\nu T\log T}}{H_{\mathcal{D},\psi}}\right).$$

It is worth noting that $H_{\mathcal{D},\psi}$ is generally a problem-dependent constant for bounded decision sets and most choices of $\psi$, as we explain in Rem. 7.

**Remark 3** (Minimal Eigenvalue Assumption). *Thm. 1 holds assuming the minimal eigenvalue of $\nabla^2\psi(\mathbf{w})$ is larger than $H_{\mathcal{D},\psi}^2$. This assumption was not required in the analysis of Scrible Abernethy et al. (2008), however, we could not circumvent it. The reason we are required to make this assumption lies in the fact the reward model we optimize is a non-linear model, whereas the reward model in Abernethy et al. (2008) is linear in $w$. This assumption is equivalent for assuming $\psi(\mathbf{w})$ is strongly convex and may hold for different choices of decision sets. For example, for a decision set which*

---

[1]Interested readers may check Luo (2017); Boyd et al. (2004); Hazan (2019) for the properties and examples of self-concordant barrier functions.

is the interior of the unit ball $\mathcal{B}_d(1)$ and choosing $\psi(\mathbf{w}) = -\ln(1 - \|\mathbf{w}\|_2^2)$ is a 1-*self concordant barrier and it is easy to check that* $H_{\mathcal{D},\psi}^2 = 2$. *Another example could be* $\psi(\mathbf{w}) = -\sum_{i=1}^d \ln w_i$, *which is a* $d$-*self-concordant barrier in the unit ball* $\mathcal{B}_d(1)$ *and it is straightforward to verify that in this case* $H_{\mathcal{D},\psi}^2 = d$.

**Remark 4** (Optimality of Thm. 1). *The rate depicted in Thm. 1 is optimal (up to logarithmic factors), as follows from the existing lower bound of the* Logit-DB *problem (Saha (2021b)).*

**Remark 5** (Advantage of Our Approach over Existing Algorithms for Logit-DB). *(1) Prior works that considered online learning in the generalized linear bandit setting Li et al. (2017; 2024) are required to assume a lower bound on the derivative of the sigmoid link function, which results in a multiplicative dependency on* $\kappa = \min_{t \in [T]} \arg\inf_{\|\boldsymbol{\theta} - \boldsymbol{\theta}_t^*\| \leq 1} \sigma'(\boldsymbol{\theta}^\top(\mathbf{x} - \mathbf{y}))$ *in the* Logit-DB *problem Saha et al. (2023); Das et al. (2024). Interestingly, we do not need to make this assumption, owing to the nice trick of exploiting the pairwise preference of symmetrically opposite points* $\mathbf{x}_t$ *and* $\mathbf{y}_t$, *as shown in Lem. 11. This is a clear advantage of our approach over the existing GLM-bandits based approach for* Logit-DB *which relies on UCB estimation based confidence bounding technique. (2) Further, since our approach relies on gradient-based techniques, they are extremely computationally efficient–the runtime requirement of our method is just* $O(dT)$, *compared to the prior methods which are computationally infeasible and not implementable in practice Saha et al. (2023); Kausik et al. (2024); Das et al. (2024).*

Due to page limitations, the complete proof of Thm. 1 is moved to App. B.

We also note that the sample complexity bound of *Double-Scrible* (Alg. 1) directly follows from Rem. 2, leading to the following result.

**Corollary 2** (Sample Complexity Bound of *Double-Scrible* (Alg. 1)). *Under the parameter settings of Thm. 1, the* $\epsilon$-*sample complexity of Double-Scrible (Alg. 1) is roughly* $O\left(\frac{d^2\nu}{H_{\mathcal{D},\psi}^2 \epsilon^2}\right)$.

**Theorem 3** (Regret Lower Bound for Logit-DB Problem). *Consider any fixed time step* $T$. *Then for any algorithm* $\mathcal{A}$ *for the* Logit-DB *problem, there exists a decision space* $\mathcal{D} \subset \mathbb{R}^d$ *and a sequence of unknown linear functionals* $\boldsymbol{\theta}_1^*, \dots \boldsymbol{\theta}_T^* \in \mathbb{R}^d$, *such that the regret of algorithm* $\mathcal{A}$ *in* $T$ *rounds* $\widehat{\mathrm{Reg}}_T^{Logit\text{-}DB} \geq \frac{d\sqrt{T \log T}}{256}$.

# 4 BATCHED FEEDBACK: ALGORITHM FOR $B$-BATCHED LOGIT-DB PROBLEM

In this section, we will analyze the batched variant of Logit-DB problem as described in Sec. 2.2. Recall that in this the learner can actively query $B$-pairwise preferences in a batched fashion.

This section can be considered a primer to our algorithm for the rank-ordered feedback setting (see Sec. 2.3) that we discussed in Sec. 5, as we will use a nice reduction of ranked-feedback setting to the batched-feedback setting.

## 4.1 ALGORITHM FOR BATCHED-LOGITDB

Our proposed algorithm for this case is *Batched-DouBle-Scrible* (*BaBle-Scrible*) which is a variant of *Double-Scrible* we detailed in the previous section for the Logit-DB. Same as algorithm, it takes input parameters $\eta$, $\delta$, and $\gamma_t$, and a $\nu$-self concordant barrier function $\psi$.

Similar to Alg. 1, in this case too the idea is to build an estimate of $\boldsymbol{\theta}_t^*$ from the pairwise observations. However, due to the batched feedback of size $B$, we can build an estimate with better variance leading to $\sqrt{B}$-factor improvement in the final learning rate of $O(\frac{d}{\sqrt{B}}\sqrt{T})$. However, one would need $B \leq d$ since it is impossible to obtain a regret rate better than $\Omega(\sqrt{T})$, which is the rate one obtains in the full information setting Zinkevich (2003).

More precisely, at any round $t$, assuming $\mathbf{w}_t$ is the running estimate of the optimizer over the decision set $\mathcal{D}_\delta$, our proposed algorithm *BaBle-Scrible* first computes the eigendecomposition of the Hessian $\nabla^2\psi(\mathbf{w}_t) = \sum_{i=1}^d \lambda_{t,i} \mathbf{v}_{t,i} \mathbf{v}_{t,i}^\top$, and samples $B$ indices $i_t^1, i_t^2, \dots, i_t^B$, uniformly from $[d]$.

Upon this it assigns $\mathbf{x}_t^\ell = \mathbf{w}_t + \gamma_t \frac{1}{2\sqrt{\lambda_{t,i_t^\ell}}} \mathbf{v}_{t,i_t^\ell}$ and $\mathbf{y}_t^\ell = \mathbf{w}_t - \gamma_t \frac{1}{2\sqrt{\lambda_{t,i_t^\ell}}} \mathbf{v}_{t,i_t^\ell}$. and plays the batch of $B$-pairs $\{(\mathbf{x}_t^1, \mathbf{y}_t^1), (\mathbf{x}_t^2, \mathbf{y}_t^2), \ldots, (\mathbf{x}_t^B, \mathbf{y}_t^B)\}$. Upon this it receives the corresponding $B$ pairwise preferences $o_t^1, \ldots, o_t^B$ and computes a gradient $(\boldsymbol{\theta}_t^*)$ estimate $\mathbf{g}_t = \frac{1}{k} \sum_{\ell=1}^{k} \mathbf{g}_t^\ell$, where $\mathbf{g}_t^\ell = \frac{d}{\gamma_t} \left( o_t^\ell - \frac{1}{2} \right) \sqrt{\lambda_{t,i_t^\ell}} \mathbf{v}_{t,i_t^\ell}$. The process is then repeated for a $T$ rounds, iteratively, refining the running estimate $\mathbf{w}_{t+1}$ by minimizing the sum of the $\psi$-regularized linearized loss over $\mathcal{D}_\delta$. Due to space limitations, the pseudocode of *BaBle-Scrible* is given in App. C.1.

Thm. 4 analyzes the regret performance of Alg. 2 which is shown to yield an optimal $O\left(\frac{d}{\min\{d,B\}}\sqrt{T}\right)$ regret for the the problem.

**Theorem 4** (Regret Analysis of Alg. 2). *Consider the decision space $\mathcal{D}$, such that $\nabla^2 \psi(\mathbf{w}) \geq H_{\mathcal{D},\psi}^2 \mathbf{I}, \ \forall \mathbf{w} \in \mathcal{D}$. Then for the choice of $\eta = \frac{\sqrt{\nu \min\{B,d\}} H_{\mathcal{D},\psi}}{d\sqrt{T \log T}}$, $\delta = \frac{1}{T}$ and $\gamma_t \leq 0.7 H_{\mathcal{D},\psi}$, the BaBle-Scrible (Alg. 1) guarantees a regret bound:*

$$\widehat{\mathrm{Reg}}_T^{\text{Batched-LogitDB}} \leq O\left( \frac{d\sqrt{\nu T \log T}}{\sqrt{\min\{B,d\}} H_{\mathcal{D},\psi}} \right).$$

The regret analysis of *BaBle-Scrible* (Alg. 2) is given in App. C.2.

Similar to Corollary 2, one can derive the sample complexity bounds for *BaBle-Scrible* (Alg. 2) using Rem. 2:

**Corollary 5** (Sample Complexity Bound of *BaBle-Scrible* (Alg. 2)). *Under the parameter settings of Thm. 4, the $\epsilon$-sample complexity of BaBle-Scrible (Alg. 2) is roughly $O\left( \frac{d^2 \nu}{\min\{d,B\} H_{\mathcal{D},\psi}^2 \epsilon^2} \right)$.*

**Theorem 6** (Regret Lower Bound for Batched `Logit-DB` Problem). *Consider any fixed time step $T$ and batched size $B$. Then for any algorithm $\mathcal{A}$ for the $B$-Batched `Logit-DB` problem, there exists a decision space $\mathcal{D} \subset \mathbb{R}^d$ and a sequence of unknown linear functionals $\boldsymbol{\theta}_1^*, \ldots \boldsymbol{\theta}_T^* \in \mathbb{R}^d$, such that the regret of algorithm $\mathcal{A}$ in $T$ rounds $\widehat{\mathrm{Reg}}_T^{\text{Batched-Logit-DB}} \geq \frac{d\sqrt{T \log T}}{256\sqrt{\min\{B,d\}}}$.*

## 5 RANKING FEEDBACK: ALGORITHM FOR MNL−BANDITS PROBLEM

In this section, we investigate the `MNL-Bandits` problem (Sec. 2.3) for the general top-$m$ ranking feedback. We first analyze the fundamental performance limit proving a lower bound for the problem. Following this we design an optimal algorithm matching the lower bound.

### 5.1 PROPOSED ALGORITHM: *MNL-Scrible*

**Useful Notations.** We will find it useful to define some notations before describing our main algorithm *MNL-Scrible*.

We denote by $V_n = \{(\pm 1)^n\}$, for any $n \in \mathbb{N}_+$. Clearly $|V_n| = 2^n$. Let $\mathcal{G}(V_n)$ be the graph with vertex set $V_n \subseteq \{\pm 1\}^n$ and there exists an (undirected) edge between two nodes $\mathbf{v}$ and $\tilde{\mathbf{v}}$ iff $\mathbf{v}$ and $\tilde{\mathbf{v}}$ only differs sign in one of the $n$ coordinates, i.e. $\exists k \in [n], \ v(k) = \tilde{v}(k)$ and $v(k') = \tilde{v}(k')$ for any $k' \neq k$. Clearly the number of neighboring nodes of any vertex $\mathbf{v} \in V_n$ in graph $\mathcal{G}$ is $|\mathcal{N}(\mathbf{v}, \mathcal{G})| = n$. In other words, the degree of any node in graph $\mathcal{G}$ is $n$. We show an example for $n = 3$ in the right figure. Also, let us define $\ell_k = \lfloor \log k \rfloor$.

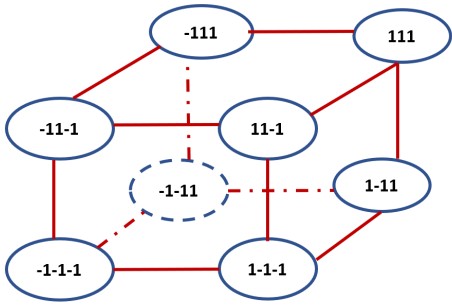

**Algorithm Description: *MNL-Scrible*** As before (in Alg. 1 and Alg. 2), this algorithm too maintains a running estimate of the minimizer $\mathbf{w}_t$ (initialized to $\mathbf{w}_1 \in \mathcal{D}$), and find the eigen decomposition of the hessian at $\mathbf{w}_t$, say $\nabla^2 \psi(\mathbf{w}_t) = \sum_{i=1}^{d} \lambda_{i_t} \mathbf{v}_{t,i} \mathbf{v}_{t,i}^\top$.

($i$). **Structured Query Sets** $S_t$**:** At each time $t$, it queries a set $S_t$ of $k$ points around $\mathbf{w}_t$ such that for every point $\mathbf{x} \in S_t$, there exists exactly $\ell_k$ neighboring points which are *symmetrically opposite to* $\mathbf{x}$ *in exactly one of the realization of* $\mathbf{v}_{t,i}s$: More precisely, at each time $t$, the algorithm first samples $\ell_k$ directions $i_t^j \sim [d]$ for $j \in [\ell_k]$ and let $U_t = \left[ \frac{\mathbf{v}_{t,i_t^1}}{\sqrt{\ell_k}}, \ldots, \frac{\mathbf{v}_{t,i_t^{\ell_k}}}{\sqrt{\ell_k}} \right] \in \mathbb{R}^{d \times \ell_k}$. We then define $S_t = \{ \mathbf{w}_t + \gamma_t U_t \mathbf{v} \mid \mathbf{v} \in V_{\ell_k} \}$.

Note that by construction indeed $S_t = 2^{\ell_k} \leq k$. Further, note for any point $\mathbf{x} = \mathbf{w}_t + \gamma U_t \mathbf{v} \in S_t$ there exists exactly $\ell_k$ symmetrically opposing points $\mathbf{x}'_i = \mathbf{w}_t + \gamma U_t \mathbf{v}'_i \in S_t$, for all $\mathbf{v}'_i \in \mathcal{N}(\mathbf{v}, \mathcal{G})$ such that $\frac{(\mathbf{x} - \mathbf{x}'_i)}{2\gamma v_i} = \mathbf{v}_{t,i_t^j}$, $j \in [\ell_k]$. Given any such point $\mathbf{x}_\mathbf{v} := \mathbf{w}_t + \gamma U_t \mathbf{v}$, let us denote by the set $\mathcal{N}(\mathbf{x}_\mathbf{v}) = \{ \mathbf{w}_t + \gamma U_t \mathbf{v}'_i \mid \mathbf{v}'_i \in \mathcal{N}(\mathbf{v}, \mathcal{G}) \}$ of all symmetrically opposing points of $\mathbf{x}$ in $S_t$ around $\mathbf{w}_t$ which differs in exactly one of the realization of $\mathbf{v}_{t,i}s$. This property will play a very crucial role in our analysis.

($ii$). **Inferring Pairwise Preferences from Top-$m$ Ranking** $\sigma_{m,t} \in \Sigma_{S_t}^m$**:** One of our critical observations is Lem. 14. Thanks to this result, we actually break the top-$m$ ranking feedback $\sigma_{m,t}$ to $m$ pairwise comparisons. In particular for any $i \in [m]$, let us denote by $\tilde{\mathbf{x}}_t^{(i)} := \mathbf{x}_t^{\sigma_{m,t}(i)}$, $S_{t \setminus i} := S_t \setminus \{ \mathbf{x}_t^{(j)} \}_{j=1}^i$, and $\mathbf{x} \succ_\sigma \mathbf{y}$ denotes $\mathbf{x}$ is preferred over $\mathbf{y}$ in the ranking $\sigma \in \Sigma_S$, for all $\mathbf{x}, \mathbf{y} \in S \subseteq \mathcal{D}$. Now note that for any $i \in [m]$, we can always find at least one $\mathbf{z} \in \mathcal{N}(\mathbf{x}_t^{(i)}) \cap \mathcal{S}_{t \setminus i}$ such that $\mathbf{x}_t^{(i)} \succ \mathbf{z}$. For ease of notation, for any such $\mathbf{x}_t^{(i)}$, we denote the corresponding rank-broken pair $\mathbf{z}$ by $\tilde{\mathbf{x}}_t^{(i)}$. Thus by definition $\mathbf{x}_t^{(i)} \succ_{\sigma_{m,t}} \tilde{\mathbf{x}}_t^{(i)}$, $\forall i \in [m]$.

($iii$). **Extracting** $m$ `Batched-LogitDB` **(batched pairwise preference) Feedback to Obtain Aggregated** $\theta_t^*$ **Estimate:** Following the notations from #($ii$) above, we extract all the pairwise comparisons $(\mathbf{x}_t^{(\ell)}, \tilde{\mathbf{x}}_t^{(\ell)})$, for all $\ell \in [m]$. Further since by definition $\tilde{\mathbf{x}}_t^{(\ell)} \in \mathcal{N}(\mathbf{x}_t^{(\ell)})$, let us denote $(\mathbf{x}_t^{(\ell)} - \tilde{\mathbf{x}}_t^{(\ell)}) = \gamma_t \mathbf{v}_t^{(\ell)}$, where note $\mathbf{v}_t^{(\ell)} = \mathbf{v}_{t,i}$ for some $i \in [d]$ and construct an aggregated gradient estimate:

$$\mathbf{g}_t = \frac{1}{B} \sum_{\ell=1}^m \mathbf{g}_t^\ell, \text{ where } \mathbf{g}_t^\ell := \frac{d}{2\gamma_t} \sqrt{\lambda_{t,i_t^{(\ell)}}} \mathbf{v}_{t,i_t^{(\ell)}}$$

($iv$). **FTRL update of** $\mathbf{w}_t$**:** Upon finding the gradient estimate $\mathbf{g}_t$, the rest of the algorithm proceeds exactly same as Alg. 1 (or Alg. 2). More precisely, The algorithm iteratively updates the decision variable $\mathbf{w}_t$ by minimizing the sum of the $\psi$-regularized linearized loss within the $\delta$-contracted decision set $\mathcal{D}_\delta$ such that $\mathbf{w}_t = \arg\min_{\mathbf{w} \in \mathcal{D}_\delta} \left\{ \eta \sum_{\tau=1}^{t-1} \mathbf{g}_\tau^\top \mathbf{w} + \psi(\mathbf{w}) \right\}$.

The complete pseudocode of *MNL-Scrible* is given in App. D.1.

**Theorem 7** (Regret Analysis of *MNL-Scrible*(Alg. 3))**.** *Consider the decision space* $\mathcal{D}$*, such that* $\nabla^2 \psi(\mathbf{w}) \geq H_{\mathcal{D},\psi}^2 \mathbf{I}$*,* $\forall \mathbf{w} \in \mathcal{D}$*.*

*Then for the choice of* $\eta = \frac{\sqrt{\nu \min\{m,d\}} H_{\mathcal{D},\psi}}{d \sqrt{T} \log T}$*,* $\delta = \frac{1}{T}$ *and* $\gamma_t \leq \frac{0.7 H_{\mathcal{D},\psi}}{\ell_k}$ *, the MNL-Scrible (Alg. 1) guarantees a regret bound:*

$$\widehat{\text{Reg}}_T^{MNL} \leq O\left( \frac{d\sqrt{\nu T \log T}}{\sqrt{m} H_{\mathcal{D},\psi}} \right).$$

The proof of Thm. 7 follows from the simple observation that upon receiving $\sigma_{m,t}$ at each round $t$, *MNL-Scrible* actually extracts $m$ independent pairwise preference feedback from $\sigma_{m,t}$. This reduces the problem to our $B$-batched pairwise feedback setting with batchsize $B = m$. The result of Thm. 7 hence follows immediately from Thm. 4.

Following the argument of Rem. 2, the sample complexity bounds of *MNL-Scrible* (Alg. 3 is as follows:

**Corollary 8** (Sample Complexity Bound of *MNL-Scrible* (Alg. 3))**.** *Under the parameter settings of Thm. 7, the $\epsilon$-sample complexity of MNL-Scrible (Alg. 3) is roughly* $O\left( \frac{d^2 \nu}{\min\{d,m\} H_{\mathcal{D},\psi}^2 \epsilon^2} \right)$*.*

**Remark 6.** *One may expect to see an improved regret rate as the number of items being simultaneously tested in each round (i.e. $k$) gets larger and larger. On the other hand, the learning*

*rate worsens since (in the worst case) it is intuitively 'harder' for the 'best-item' of a $k$-subset to prove its supremacy against the $k-1$-competitors due to higher outcome variance. The result, in a sense, formally establishes that the former advantage is nullified by the latter drawback yielding a $k$-independent guarantee. One really needs to consider a worst-case problem instance for this interplay to happen, as we carefully construct it in our lower bound derivation of Thm. 9.*

**Theorem 9** (Regret Lower Bound for Top-$m$ Ranking `MNL-Bandits` Problem). *Consider any fixed time step $T$, subsetsize $k$ and length of rank-ordered feedback $m \in \{1, \ldots, k\}$. Then for any algorithm $\mathcal{A}$ for the Top-$m$ `MNL-Bandits` problem, there exists a decision space $\mathcal{D} \subset \mathbb{R}^d$ and a sequence of unknown linear functionals $\boldsymbol{\theta}_1^*, \ldots \boldsymbol{\theta}_T^* \in \mathbb{R}^d$, such that the regret of algorithm $\mathcal{A}$ in $T$ rounds $\widehat{\mathrm{Reg}}_T^{MNL} \geq \frac{d\sqrt{T \log T}}{256\sqrt{\min\{m,d\}}}$.*

## 6 EXPERIMENTS

We run synthetic experiments to report the performance of our methods, *Double-Scrible* (Sec. 3) and *MNL-Scrible* (Sec. 5) respectively, for dueling and top-$m$ ranking feedback. All results are averaged across 100 runs. We run our experiments on different environments (adversarial loss sequences): Precisely, for a fixed $d$, we construct $\theta_t^* = \mathbf{1} + \varepsilon$

**Adversarial ($\boldsymbol{\theta}$) Environments.** We report our experiment results on problem instances with varying dimension $d$ generated as follows: (1) Inst-1: For a given $d$ and round $t$, we choose $\theta_t$ (2) Inst-2: For a given $d$ and round $t$, we choose $\theta_t$

The decision space is given by $\mathcal{D} = \{\mathbf{x} \mid \mathbf{Ax} \leq \mathbf{b}\}$, for some $A \in \mathbb{R}^{c \times d}$ and $\mathbf{b} \in \mathbb{R}^c$, for some $c \in \mathbb{N}_+$.

**Choice of the self concordant barrier $\psi$:** We use the following self concordant barrier $\psi(w) = -\sum_{j=1}^m \ln(b_j - \mathbf{a}_j^\top w)$, which is known to be an $c$-self-concordant barrier for $\mathcal{D}$, $\mathbf{a}_j$ being the $j$-th row of $\mathbf{A}$.

### 6.1 WITH VARYING $d$

Our first experiments are reported for the Adversarial Logistic Dueling Bandits (`Logit-DB`) setting on Inst-1 and

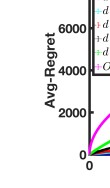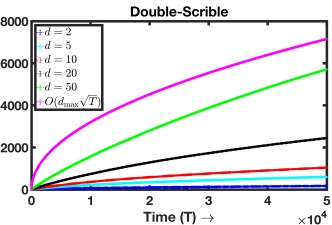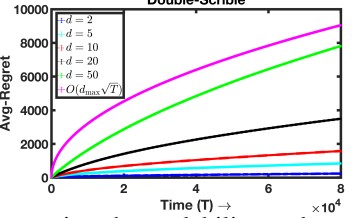

Inst-2. We run the experiments for $d = 2, 5, 10, 20, 50$ to examine the scalability and runtime efficiency of our proposed methods which are provably shown to scale as $O(d\sqrt{T})$.

### 6.2 RUNTIME COMPARISON.

We now report the (averaged) run-times of the above executions in seconds. Note the first experiment is run for $T = 50,000$ while the second experiment is run for $T = 80,000$ rounds.

| d | runtime (sec) |
|---|---|
| 2 | 1.459960 |
| 6 | 1.773056 |
| 10 | 2.149256 |
| 20 | 3.150031 |
| 50 | 11.567123 |

T = 50,000 (Inst-1)

| d | runtime (sec) |
|---|---|
| 2 | 2.339139 |
| 6 | 3.035577 |
| 10 | 3.731622 |
| 20 | 5.036075 |
| 40 | 17.398917 |

T = 80,000 (Inst-2)

### 6.3 PERFORMANCE OF *MNL-Scrible*: WITH VARYING $m$

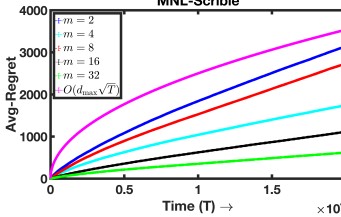

For top-$m$ ranking setting, we used $m = 2, 4, 8, 16, 32$ for $d = 50$ on Inst-1.

## 7 CONCLUSION

In this paper, we introduced an efficient gradient descent-based approach for regret minimization for online linear optimization with adversarial preferences. Our results have critical implications in learning problems of RLHF which has wide applications in fields of AI-alignment, fine-tuning language models, etc. Our proposed novel online mirror descent (OMD) algorithm achieves an optimal regret bound of $O(\sqrt{T})$ while only relying on binary preference feedback. This advancement improves upon existing methods by addressing key computational challenges, particularly in handling high-dimensional and adversarial environments while still respecting optimal performance guarantees. We also extended our algorithm to accommodate $B$-batched preference feedback and $m$-partial ranking on $k$-subsets, which is shown to yield improved performance guarantees depending on the batch size $B$ or the length of the rank-ordered feedback $m$. The computational efficiency of our algorithms makes them suitable for large-scale real-world applications.

**Future Work.** Building on this work, several promising avenues for future exploration emerge: One potential extension is to generalize the setting beyond linear scores which is certainly not straightforward even for value-feedback based convex optimization setting . Extending to partially observable preferences or partial ranking feedback over a subset of alternatives is also an interesting open problem. Another direction is to explore hybrid approaches that combine gradient descent with other optimization techniques like Thompson sampling or Bayesian methods, to reduce variance in feedback-based learning. Finally, investigating how this algorithm can be adapted for different AI alignment challenges, such as incorporating fairness or ethical constraints in decision-making, presents an exciting opportunity for future research.

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
