{\mathrm{Reg}}_T^{\text{Logit-DB}}$ would suffice to bound $\mathrm{Reg}_T^{\text{Logit-DB}}$.*

*Proof of Rem. 1.* Let us fix a round $t$, and for simplicity denote $\mathbf{x}^* = \mathbf{x}_t^*$ (dropping the subscript). Note that due to the underlying preference structure for any $\mathbf{x} \in \mathcal{D}$,

$$P_t(\mathbf{x}^*, \mathbf{x}) - 1/2 = \sigma(\boldsymbol{\theta}_t^{*\top}(\mathbf{x}^* - \mathbf{x})) - 1/2 = \frac{\exp(\boldsymbol{\theta}_t^{*\top}\mathbf{x}^*)}{\exp(\boldsymbol{\theta}_t^{*\top}\mathbf{x}^*) + \exp(\boldsymbol{\theta}_t^{*\top}\mathbf{x})} - 1/2$$

$$= \frac{\left(\exp(\boldsymbol{\theta}_t^{*\top}(\mathbf{x}^* - \mathbf{x})) - 1)\right)}{2\left(\exp(\boldsymbol{\theta}_t^{*\top}(\mathbf{x}^* - \mathbf{x})) + 1)\right)} \overset{(a)}{\geq} \frac{\left(\exp(\boldsymbol{\theta}_t^{*\top}(\mathbf{x}^* - \mathbf{x})) - 1)\right)}{4}$$

$$= \frac{1}{4}\left(1 + \sum_{i=1}^\infty \frac{(\boldsymbol{\theta}_t^{*\top}(\mathbf{x}^* - \mathbf{x}))^i}{i!} - 1\right) > \frac{\boldsymbol{\theta}_t^{*\top}(\mathbf{x}^* - \mathbf{x})}{4}$$

where (a) follows since $\boldsymbol{\theta}_t^{*\top}\mathbf{x} \in [0, 1]$, $\forall \mathbf{x} \in \mathcal{D}$, assuming $\boldsymbol{\theta}_t^* \in \mathcal{B}_d(1)$ and $\mathcal{D} \subseteq \mathcal{B}_d(1)$. On the other hand,

$$P_t(\mathbf{x}^*, \mathbf{x}) - 1/2 = \sigma(\boldsymbol{\theta}_t^{*\top}(\mathbf{x}^* - \mathbf{x})) - 1/2 = \frac{\exp(\boldsymbol{\theta}_t^{*\top}\mathbf{x}^*)}{\exp(\boldsymbol{\theta}_t^{*\top}\mathbf{x}^*) + \exp(\boldsymbol{\theta}_t^{*\top}\mathbf{x})} - 1/2$$

$$= \frac{\left(\exp(\boldsymbol{\theta}_t^{*\top}(\mathbf{x}^* - \mathbf{x})) - 1)\right)}{2\left(\exp(\boldsymbol{\theta}_t^{*\top}(\mathbf{x}^* - \mathbf{x})) + 1)\right)} \overset{(b)}{\leq} \frac{\left(\exp(\boldsymbol{\theta}_t^{*\top}(\mathbf{x}^* - \mathbf{x})) - 1)\right)}{2}$$

$$= \frac{1}{2}\left(1 + \sum_{i=1}^\infty \frac{(\boldsymbol{\theta}_t^{*\top}(\mathbf{x}^* - \mathbf{x}))^i}{i!} - 1\right).$$

Now let us denote $a = \boldsymbol{\theta}_t^{*\top}(\mathbf{x}^* - \mathbf{x})$ and note by assumption $0 < a \leq 2$.

$$P_t(\mathbf{x}^*, \mathbf{x}) - 1/2 = \frac{1}{2}\left(\sum_{i=1}^\infty \frac{a^i}{i!}\right) \leq \frac{a}{2} + \frac{a^2}{2}\left(1 + \frac{a}{2} + \frac{a^2}{2^2} + \dots\right)$$

$$= \frac{a}{2} + \frac{a^2}{2}\frac{2}{(2-a)} \leq \frac{a}{2} + \frac{a}{2} = a = \boldsymbol{\theta}_t^{*\top}(\mathbf{x}^* - \mathbf{x}),$$

where the second the last inequality holds since $a \in (0, 2)$, assuming $\boldsymbol{\theta}_t^* \in \mathcal{B}_d(1)$ and $\mathcal{D} \subseteq \mathcal{B}_d(1)$.
$\square$

# B APPENDIX FOR SEC. 3

## B.1 *Double-Scrible*: ALGORITHM PSEUDOCODE

---
**Algorithm 1** *Double-Scrible*

---
1: Input: Decision set $\mathcal{D}$ with $\nu$-self concordant barrier $\psi$, parameters $\eta, \delta, \gamma_t$.
2: **for** $t = 1$ to $T$ **do**
3:     Compute: $\mathbf{w}_t = \arg\min_{\mathbf{w} \in \mathcal{D}_\delta} \left\{ \eta \sum_{\tau=1}^{t-1} (-\mathbf{g}_\tau)^\top \mathbf{w} + \psi(\mathbf{w}) \right\}$.
4:     Compute eigendecomposition s.t. $\nabla^2 \psi(\mathbf{w}_t) = \sum_{j=1}^d \lambda_{t,j} \mathbf{v}_{t,j} \mathbf{v}_{t,j}^\top$.
5:     Sample $i_t \in [d]$ uniformly at random
6:     Choose $\mathbf{x}_t = \mathbf{w}_t + \gamma_t \frac{1}{2\sqrt{\lambda_{t,i_t}}} \mathbf{v}_{t,i_t}$ and $\mathbf{y}_t = \mathbf{w}_t - \gamma_t \frac{1}{2\sqrt{\lambda_{t,i_t}}} \mathbf{v}_{t,i_t}$.
7:     Play $(\mathbf{x}_t, \mathbf{y}_t)$, observe $o_t \sim \text{Ber}(P_t(\mathbf{x}_t, \mathbf{y}_t))$.
8:     $\mathbf{g}_t = \frac{d}{\gamma_t} \left(o_t - \frac{1}{2}\right) \sqrt{\lambda_{t,i_t}} \mathbf{v}_{t,i_t}$.
9: **end for**

---

## B.2 KEY LEMMAS FOR THM. 1 (REGRET ANALYSIS OF ALG. 1)

We define the useful notations which will be useful for stating the claims:

**Notations:** We denote the history $\mathcal{H}_t := \{(i_1, o_1), (i_2, o_2), \ldots (i_{t-1}, o_{t-1})\}$ till time $t$. We define a norm associated with the Hessian of $\psi$ at $\mathbf{w}$ as $\|\mathbf{x}\|_{\mathbf{w}} = \|\mathbf{x}\|_{\nabla^2 \psi(\mathbf{w})} = \sqrt{\mathbf{x}^\top \nabla^2 \psi(\mathbf{w}) \mathbf{x}}$ for any $\mathbf{x} \in \mathbb{R}^d$. This is indeed a norm since a self-concordant barrier is strictly convex, such that $\nabla^2 \psi(\mathbf{w})$ is positive definite for any $\mathbf{w} \in \text{int}(\mathcal{D})$.

Further considering the eigen-decomposition of $\nabla^2 \psi(\mathbf{w}) = \sum_{i=1}^d \lambda_i \mathbf{v}_i \mathbf{v}_i^\top$, we further note that

$$\|\mathbf{x}\|_{\mathbf{w}} = \sqrt{\mathbf{x}^\top \nabla \psi(\mathbf{w}) \mathbf{x}} = \sqrt{\sum_{i=1}^d \lambda_{t,i} \mathbf{x}^\top (\mathbf{v}_{t,i} \mathbf{v}_{t,i}^\top) \mathbf{x}} = \sqrt{\sum_{i=1}^d \lambda_i (\mathbf{x}^\top \mathbf{v}_{t,i})^2}, \ \forall \mathbf{x} \in \mathbb{R}^d.$$

Further, one can define the dual norm of Hessian of $\psi$ at $\mathbf{w}$ as:

$$\|\mathbf{x}\|_{\mathbf{w}}^* = \sqrt{\mathbf{x}^\top \nabla^{-2} \psi(\mathbf{w}) \mathbf{x}} = \sqrt{\sum_{i=1}^d \frac{1}{\lambda_{t,i}} \mathbf{x}^\top (\mathbf{v}_{t,i} \mathbf{v}_{t,i}^\top) \mathbf{x}}, \ \forall \mathbf{x} \in \mathbb{R}^d.$$

The Dikin ellipsoid centered at $\mathbf{w}$ with radius $r$ is defined as the ellipsoid

$$\mathcal{E}_r(\mathbf{w}) = \left\{ \mathbf{x} \in \mathbb{R}^d : \|\mathbf{x} - \mathbf{w}\|_{\mathbf{w}} \le r \right\}.$$

**Property 1** (Luo (2017); Boyd et al. (2004)). *If $\psi$ is a self-concordant barrier on $\mathcal{D}$, then $\mathcal{E}_1(\mathbf{w}) \subset \mathcal{D}$ for any $\mathbf{w} \in \text{int}(\mathcal{D})$.*

**Property 2** (Luo (2017)). *Let $\mathbf{x} \in \text{int}(\mathcal{D})$ be such that $\|\nabla \Phi(\mathbf{x})\|_{\mathbf{x}}^* \le \frac{1}{4}$, and let $\mathbf{x}^\star = \arg\min_{\mathbf{x} \in \mathcal{D}} \Phi(\mathbf{x})$. Then for any $\Phi : \mathcal{D} \mapsto \mathbb{R}$,*

$$\|\mathbf{x} - \mathbf{x}^\star\|_{\mathbf{x}} \le 2\|\nabla \Phi(\mathbf{x})\|_{\mathbf{x}}^*.$$

**Property 3** (Luo (2017); Hazan (2019)). *Let $\psi$ be a $\nu$-self concordant function over $\mathcal{D}$, then for all $\mathbf{x}, \mathbf{y} \in \text{int}(\mathcal{D})$:*

$$\psi(\mathbf{y}) - \psi(\mathbf{x}) \le \nu \log \frac{1}{1 - \pi_{\mathbf{x}}(\mathbf{y})},$$

*where $\pi_{\mathbf{x}}(\mathbf{y}) = \inf\{t \ge 0 : \mathbf{x} + t^{-1}(\mathbf{y} - \mathbf{x}) \in \mathcal{D}\}$.*

The proof sketch of Thm. 1 depends on some key lemmas. First we claim that $g_t$ given an 'almost' unbiased estimate of $\boldsymbol{\theta}_t^*$ up to some constants.

**Lemma 10** (Ensuring Decision Boundaries). *At any round $t$, $\mathbf{x}_t$ and $\mathbf{y}_t \in \mathcal{D}$ in Alg. 1.*

*Proof.* We will prove the result for $\mathbf{x}_t$. A similar analysis will apply to $\mathbf{y}_t$ as well. Note since $\mathbf{w}_t \in \text{int}(\mathcal{D})$, and $\|\mathbf{x}_t - \mathbf{w}_t\|_{\mathbf{x}} \leq \gamma_t \leq 1$. Note Rem. 7 ensures $\gamma_t \leq 1$ and thus the results follows using Property 1. $\qquad\square$

**Lemma 11** (Gradient Estimation). *It can be shown that for any round $t$,*

$$\mathbf{E}[\mathbf{g}_t \mid \mathcal{H}_t] = C\boldsymbol{\theta}_t^*,$$

*for some $C \in [0.22, 0.25]$, whenever $\gamma_t \leq 0.7\sqrt{\lambda_{\min}(\nabla^2\psi(\mathbf{w}_t))}$.*

**Remark 7** (Ensuring appropriate choice of $\gamma_t$). *Noting that, given the decision space $\mathcal{D}$, since $\nabla^2\psi(\mathbf{w}_t) \geq H_{\mathcal{D},\psi}^2 \mathbf{I}_d$, one can easily satisfy $\gamma_t \leq 0.7\sqrt{\lambda_{\min}(\nabla^2\psi(\mathbf{w}_t))}$ by choosing $\gamma_t = \min\{1, 0.7 H_{\mathcal{D},\psi}\}$. We have given some specific examples in Rem. 3.*

*Proof.* Consider any fixed round $t \in [T]$. We note that:

$$\mathbf{E}_{o_t}[(o_t - 1/2) \mid i_t, \mathcal{H}_t] = \mathbf{E}_{o_t}[o_t \mid i_t, \mathcal{H}_t] - 1/2 = \sigma\big(\boldsymbol{\theta}_t^{*\top}(\mathbf{x}_t - \mathbf{y}_t)\big) - 1/2$$

$$= \sigma\big((2\gamma_t/\sqrt{\lambda_{t,i_t}})\boldsymbol{\theta}_t^{*\top}\mathbf{v}_{t,i_t}\big) - 1/2$$

$$= \sigma'(\varepsilon_t)(\gamma_t/\sqrt{\lambda_{t,i_t}})\boldsymbol{\theta}_t^{*\top}\mathbf{v}_{t,i_t} \quad [\text{using MVT, where } |\varepsilon_t| \in [0, |(\gamma_t/\sqrt{\lambda_{t,i_t}})\boldsymbol{\theta}_t^{*\top}\mathbf{v}_{t,i_t}|]. \quad (1)$$

Let us denote $c_t = |(\gamma_t/\sqrt{\lambda_{t,i_t}})\boldsymbol{\theta}_t^{*\top}\mathbf{v}_{t,i_t}|$ and note that we can bound $c_t \leq \frac{\gamma_t}{\sqrt{\lambda_{t,i_t}}}\|\boldsymbol{\theta}_t^*\|\|\mathbf{v}_{t,i_t}\| \leq \frac{\gamma_t}{\sqrt{\lambda_{\min}(\nabla^2\psi(\mathbf{w}_t))}}$, where the first inequality follows from the Cauchy-Schwarz inequality.

Then by choosing any $\gamma_t \leq 0.7\sqrt{\lambda_{\min}(\nabla^2\psi(\mathbf{w}_t))}$ we get $c_t \leq 0.7$. This along with the results of Lem. 15 (App. E) implies that $\sigma'(\epsilon_t) \leq [0.222, 0.25]$ for the appropriate choice of $\gamma_t$. Note Rem. 7 explains the suitable choice of $\gamma_t$ For simplicity, we will use $L = 0.222$, $U = 0.25$ for the rest of this proof and let $\sigma'(\varepsilon_t) \in [L, U]$. The interesting thing now is to note that, given the history $\mathcal{H}_t$ till time $t$, $\mathbf{g}_t$ in Alg. 1 satisfies:

$$\mathbf{E}_{o_t,i_t}[g_t \mid \mathcal{H}_t] = \mathbf{E}_{i_t,\omega_t}\left[\frac{d}{\gamma_t}\mathbf{E}_{o_t}\left[\left(o_t - \frac{1}{2}\right) \mid i_t, \mathcal{H}_t\right]\sqrt{\lambda_{t,i_t}}\mathbf{v}_{t,i_t}\right]$$

$$= \mathbf{E}_{i_t}\left[\frac{d}{\gamma_t}\left(\sigma'(\varepsilon_t)(\gamma_t/\sqrt{\lambda_{t,i_t}})\boldsymbol{\theta}_t^{*\top}\mathbf{v}_{t,i_t}\right)\sqrt{\lambda_{t,i_t}}\mathbf{v}_{t,i_t}\right] \quad \text{using Eq. (1)}$$

$$\in \left[L\mathbf{E}_{i_t}\left[\frac{d}{\gamma_t}\left((\gamma_t/\sqrt{\lambda_{t,i_t}})\boldsymbol{\theta}_t^{*\top}\mathbf{v}_{t,i_t}\right)\sqrt{\lambda_{t,i_t}}\mathbf{v}_{t,i_t}\right], U\mathbf{E}_{i_t}\left[\frac{d}{\gamma_t}\left((\gamma_t/\sqrt{\lambda_{t,i_t}})\boldsymbol{\theta}_t^{*\top}\mathbf{v}_{t,i_t}\right)\sqrt{\lambda_{t,i_t}}\mathbf{v}_{t,i_t}\right]\right]$$

$$\in [L\boldsymbol{\theta}_t^*, U\boldsymbol{\theta}_t^*],$$

where the last inequality follows noting:

$$\mathbf{E}_{i_t}\left[\frac{d}{\gamma_t}\left((\gamma_t/\sqrt{\lambda_{t,i_t}})\boldsymbol{\theta}_t^{*\top}\mathbf{v}_{t,i_t}\right)\sqrt{\lambda_{t,i_t}}\mathbf{v}_{t,i_t}\right] = \mathbf{E}_{i_t}\left[d\left((1/\sqrt{\lambda_{t,i_t}})\boldsymbol{\theta}_t^{*\top}\mathbf{v}_{t,i_t}\right)\sqrt{\lambda_{t,i_t}}\mathbf{v}_{t,i_t}\right]$$

$$= d\left(\sum_{i=1}^d \frac{1}{d\sqrt{\lambda_{t,i}}}\sqrt{\lambda_{t,i}}\mathbf{v}_{t,i}\mathbf{v}_{t,i}^\top\right)\boldsymbol{\theta}_t^* = \boldsymbol{\theta}_t^*,$$

since $\sum_i \mathbf{v}_{t,i}\mathbf{v}_{t,i}^\top = \mathbf{I}_d$ by the fact that $\{\mathbf{v}_{t,i}\}_{i\in[d]}$ are orthonormal vectors that span $\mathbb{R}^d$. $\qquad\square$

Equipped with the previous results, we are now ready to proof our main theorem, Thm. 1, as shown below.

### B.3 REGRET ANALYSIS: PROOF OF THM. 1

Suppose *Be The Leader* (BTL) algorithm Cesa-Bianchi & Lugosi (2006a); Ma (2018) is run on the loss vector sequence $-\mathbf{g}_1, -\mathbf{g}_2, \ldots, -\mathbf{g}_T)$, $\mathbf{g}_i \in \mathbb{R}^d$. We know that for any $\mathbf{u} \in \mathcal{D}_\delta$:

$$\sum_{t=1}^{T} (\mathbf{w}_t - \mathbf{u})^\top (-\mathbf{g}_t) \le \sum_{t=1}^{T} (\mathbf{w}_t - \mathbf{w}_{t+1})^\top (-\mathbf{g}_t) + \frac{(\psi(\mathbf{u}) - \psi(\mathbf{w}_1))}{\eta}.$$

Further applying Holder's inequality, we get:

$$\sum_{t=1}^{T} (\mathbf{u} - \mathbf{w}_t)^\top \mathbf{g}_t \le \sum_{t=1}^{T} \|\mathbf{w}_t - \mathbf{w}_{t+1}\|_{\mathbf{w}_t} \| -\mathbf{g}_t \|_{\mathbf{w}_t}^* + \frac{(\psi(\mathbf{u}) - \psi(\mathbf{w}_1))}{\eta}. \tag{2}$$

Note we defined: $\mathbf{g}_t = \frac{d}{\gamma_t}\left(o_t - \frac{1}{2}\right)\sqrt{\lambda_{t,i_t}}\mathbf{v}_{t,i_t}$ and by Lem. 11, we have

$$0.22\boldsymbol{\theta}_t^* \le \mathbf{E}[g_t \mid \mathcal{H}_t] \le 0.25\boldsymbol{\theta}_t^*,$$

which implies :

$$\boldsymbol{\theta}_t^{*\top}(\mathbf{u} - \mathbf{w}_t) \le \frac{1}{0.22}\mathbf{E}[g_t^\top \mid \mathcal{H}_t](\mathbf{u} - \mathbf{w}_t), \tag{3}$$

combining this with Eq. (2), we get:

$$0.22\boldsymbol{\theta}_t^{*\top}(\mathbf{u} - \mathbf{w}_t) \le \sum_{t=1}^{T} \|\mathbf{w}_t - \mathbf{w}_{t+1}\|_{\mathbf{w}_t} \|\mathbf{g}_t\|_{\mathbf{w}_t}^* + \frac{(\psi(\mathbf{u}) - \psi(\mathbf{w}_1))}{\eta}. \tag{4}$$

On the other hand, by definition of $\|\cdot\|_{\mathbf{w}_t}^*$, we have that for any realization of $\mathbf{g}_t$:

$$\|\mathbf{g}_t\|_{\mathbf{w}_t}^* = \sqrt{\sum_{i=1}^{d} \frac{1}{\lambda_{i,t}}\mathbf{g}_t^\top(\mathbf{v}_{t,i}\mathbf{v}_{t,i}^\top)\mathbf{g}_t} = \frac{d}{2\gamma_t}. \tag{5}$$

Additionally, let us denote by $\Phi_t(\mathbf{w}) = \eta\sum_{\tau=1}^{t}(-\mathbf{g}_\tau)^\top \mathbf{w} + \psi(\mathbf{w})$, then note Alg. 1 have $\mathbf{w}_{t+1} = \arg\min_{\mathbf{w}\in\mathcal{D}_\delta} \Phi_t(x)$. Thus, applying Property 2, we get:

$$\|\mathbf{w}_t - \mathbf{w}_{t+1}\|_{\mathbf{w}_t} \le 2\|\nabla\Phi_t(\mathbf{w}_t)\|_{\mathbf{w}_t}^* = 2\|\nabla\Phi_{t-1}(\mathbf{w}_t) + \eta\mathbf{g}_t\|_{\mathbf{w}_t}^* = 2\eta\|\mathbf{g}_t\|_{\mathbf{w}_t}^*,$$

where note by definition $\nabla\Phi_{t-1}(\mathbf{w}_t) = 0$ by definition of $\mathbf{w}_t$ for all $t$. But note for Property 2 to be applied we need $\|\nabla\Phi_t(\mathbf{w}_t)\|_t^* \le \frac{1}{4}$, but this is indeed true since since Eq. (7) implies

$$\eta\|\mathbf{g}_t\|_{\mathbf{w}_t}^* \le \frac{\eta d}{2\gamma_t},$$

and thus choosing any $\eta \le \frac{\gamma_t}{2d}$, we have $\|\nabla\Phi_t(\mathbf{w}_t)\|_t^* \le \frac{1}{4}$, as desired. We will see shortly how to choose $\eta$ to ensure $\eta \le \frac{\gamma_t}{2d}$.

Further using Property 2, we have $\|\mathbf{w}_t - \mathbf{w}_{t+1}\|_{\mathbf{w}_t} \le 2\eta\|\mathbf{g}_t\|_{\mathbf{w}_t}^*$, which along with Eq. (4) we get:

$$0.22\sum_{t=1}^{T}(\mathbf{u} - \mathbf{w}_t)^\top \boldsymbol{\theta}_t^* \le \sum_{t=1}^{T} 2\eta\|\mathbf{g}_t\|_{\mathbf{w}_t}^{*2} + \frac{(\psi(\mathbf{u}) - \psi(\mathbf{w}_1))}{\eta}$$

$$= 2\eta\sum_{t=1}^{T} \frac{d^2}{4\gamma_t^2} + \frac{\nu\log\frac{1}{1-\pi_{\mathbf{w}_1}(\mathbf{u})}}{\eta}.$$

However noting $\mathbf{u}$ and $\mathbf{w}_1 \in \mathcal{D}_\delta$ , by definition of $\pi_{\mathbf{w}_1}(u) = (1 - \delta)$ in Property 3, implying:

$$0.22 \sum_{t=1}^{T} \left( \mathbf{u} - \mathbf{w}_t \right)^\top \boldsymbol{\theta}_t^* \leq \eta \sum_{t=1}^{T} \frac{d^2}{2\gamma_t^2} + \frac{\nu \log \frac{1}{\delta}}{\eta}. \tag{6}$$

Further if we choose $\mathbf{u} := \arg\max_{\mathbf{x} \in \mathcal{D}_\delta} \sum_{t=1}^{T} \boldsymbol{\theta}_t^{*\top} \mathbf{x}$, and recalling that we defined $\mathbf{x}^* := \arg\max_{\mathbf{x} \in \mathcal{D}} \sum_{t=1}^{T} \boldsymbol{\theta}_t^{*\top} \mathbf{x}$, note that:

$$\sum_{t=1}^{T} \left( \mathbf{x}^* - \mathbf{w}_t \right)^\top \boldsymbol{\theta}_t^* \leq \sum_{t=1}^{T} \left( u - \mathbf{w}_t \right)^\top \boldsymbol{\theta}_t^* + \delta T L D$$

$$= \frac{1}{0.22} \left[ \eta \sum_{t=1}^{T} \frac{d^2}{2\gamma_t^2} + \frac{\nu \log \frac{1}{\delta}}{\eta} \right] + \delta T L D, \qquad \text{from Eq. (6)}$$

$$\leq \frac{1}{0.22} \left[ \frac{\eta d^2 T}{\min\{1, H_{\mathcal{D},\psi}^2\}} + \frac{\nu \log \frac{1}{\delta}}{\eta} \right] + \delta T L D, \quad \text{since we chose } \gamma \leq \min\{1, 0.7 H_{\mathcal{D},\psi}\}$$

$$= \frac{d \sqrt{\nu T \log T}}{0.22 H_{\mathcal{D},\psi}} + L D,$$

choosing $\eta = \frac{\sqrt{\nu} H_{\mathcal{D},\psi}}{d \sqrt{T \log T}}$ and $\delta = \frac{1}{T}$, concludes the prove noting the diameter of the decision set $\mathcal{D} \subseteq \mathcal{B}_d(1)$ is bounded by 1, and the lipschitz constant $L \leq \max_{t \in [T]} \|\boldsymbol{\theta}_t^*\| \leq 1$.

## C APPENDIX FOR SEC. 4

### C.1 *BaBle-Scrible*: ALGORITHM PSEUDOCODE

---

**Algorithm 2** *BaBle-Scrible*

---

1: Input: Decision set $\mathcal{D}$ with $\nu$-self concordant barrier $\psi$, parameters $\eta, \delta, \gamma_t$.
2: **for** $t = 1$ to $T$ **do**
3:     Compute: $\mathbf{w}_t = \arg\min_{\mathbf{w} \in \mathcal{D}_\delta} \left\{ \eta \sum_{\tau=1}^{t-1} (-\mathbf{g}_\tau)^\top \mathbf{w} + \psi(\mathbf{w}) \right\}.$
4:     Compute eigendecomposition s.t. $\nabla^2 \psi(\mathbf{w}_t) = \sum_{i=1}^{d} \lambda_{t,i} \mathbf{v}_{t,i} \mathbf{v}_{t,i}^\top.$
5:     **for** $\ell = 1, 2, \ldots, B$ **do**
6:         Sample $i_t^\ell \in [d]$ uniformly at random.
7:         Choose $\mathbf{x}_t^\ell = \mathbf{w}_t + \gamma_t \frac{1}{2\sqrt{\lambda_{t,i_t^\ell}}} \mathbf{v}_{t,i_t^\ell}$ and $\mathbf{y}_t^\ell = \mathbf{w}_t - \gamma_t \frac{1}{2\sqrt{\lambda_{t,i_t^\ell}}} \mathbf{v}_{t,i_t^\ell}.$
8:         Play $(\mathbf{x}_t^\ell, \mathbf{y}_t^\ell)$, observe $o_t^\ell \sim \text{Ber}(P_t(\mathbf{x}_t^\ell, \mathbf{y}_t^\ell)).$
9:         $\mathbf{g}_t^\ell = \frac{d}{\gamma_t} \left( o_t^\ell - \frac{1}{2} \right) \sqrt{\lambda_{t,i_t^\ell}} \mathbf{v}_{t,i_t^\ell}.$
10:    **end for**
11:    Update $\mathbf{g}_t = \frac{1}{B} \sum_{\ell=1}^{B} \mathbf{g}_t^\ell$
12: **end for**

---

### C.2 REGRET ANALYSIS OF ALG. 2

We will need to prove some key lemmas before proceeding to the proof of the main theorem $Thm.$ 4.

**Lemma 12** (Gradient ($\boldsymbol{\theta}_t^*$) Estimation)**.** *It can be shown that for any round t,*

$$\mathbf{E}[g_t \mid \mathcal{H}_t] = C \boldsymbol{\theta}_t^*,$$

*for some $C \in [0.22, 0.25]$, whenever $\gamma_t \leq 0.7 \sqrt{\lambda_{\min}(\nabla^2 \psi(\mathbf{w}_t))}$.*

*Proof of Lem. 12.* Let us fix any $t \in [T]$. Recall that we defined $\mathbf{g}_t^\ell = \frac{d}{\gamma_t}\left(o_t^\ell - \frac{1}{2}\right)\sqrt{\lambda_{t,i_t^\ell}}\mathbf{v}_{t,i_t^\ell}$ and for any $\ell = 1, 2, \ldots, B$, Now noting since $i_t^\ell \sim \text{Unif}([d])$, following the notations and exact same proof of Lem. 11, we get that: for any $\ell \in [B]$, $\mathbf{E}[g_t^\ell \mid \mathcal{H}_t] = C\boldsymbol{\theta}_t^*$, for some $C \in [0.22, 0.25]$. The proof now follows noting $\mathbf{g}_t := \frac{1}{B}\sum_{\ell=1}^B \mathbf{g}_t^\ell$. $\qquad\square$

We next prove the most important claim of this analysis that shows that indeed the batched feedback helped to obtain a more accurate (reduced variance) estimate of the gradient $\boldsymbol{\theta}_t^*$ at each time step $t$. The proof involves a smart exploitation of the second moment of Binomial distribution, we will see in the proof of Lem. 13.

**Lemma 13** (Improved Variance of $\mathbf{g}_t$ (Norm bound)). *At any time $t$, one can show that*

$$\mathbf{E}_{i_t^\ell, o_t^\ell}[\|\mathbf{g}_t\|_{\mathbf{w}_t}^*] \leq \frac{d}{\gamma_t\sqrt{\{B, d\}}}. \tag{7}$$

*Proof of Lem. 13.* We start by recalling that we defined the dual norm of Hessian of $\psi$ at $\mathbf{w}$ as

$$\|\mathbf{x}\|_{\mathbf{w}}^* = \sqrt{\mathbf{x}^\top \nabla^{-2}\psi(\mathbf{w})\mathbf{x}} = \sqrt{\sum_{i=1}^d \frac{1}{\lambda_{t,i}}\mathbf{x}^\top(\mathbf{v}_{t,i}\mathbf{v}_{t,i}^\top)\mathbf{x}}, \ \ \forall \mathbf{x} \in \mathbb{R}^d.$$

At any round $t$, let us now denote by $N_{t,i}$ the number of times the $i$-th eigen basis, $\mathbf{v}_{t,i}$, was drawn at round $t$, $i \in [d]$. Clearly $\sum_{i=1}^d N_{t,i} = B$. With this view we note that:

$$\mathbf{g}_t = \frac{1}{B}\sum_{\ell=1}^B \mathbf{g}_t^\ell = \frac{d}{B\gamma_t}\sum_{i=1}^d N_{t,i}\left(o_t^i - \frac{1}{2}\right)\sqrt{\lambda_{t,i}}\mathbf{v}_{t,i},$$

and noting that since $\mathbf{v}_i$s are orthogonal to each other:

$$\mathbf{E}_{i_t^1, o_t^1, \ldots i_t^d, o_t^d}\left[\|\mathbf{g}_t\|_{\mathbf{w}_t}^*\right] \leq \frac{d}{2B\gamma_t}\mathbf{E}_{i_t^1, \ldots, i_t^d}\left[\sqrt{\sum_{i=1}^d N_{t,i}^2 \mathbf{v}_{t,i}^\top(\mathbf{v}_{t,i}\mathbf{v}_{t,i}^\top)\mathbf{v}_{t,i}}\right]$$

$$= \frac{d}{2B\gamma_t}\sqrt{\sum_{i=1}^d \mathbf{E}_{i_t}\left[N_{t,i}^2\right]}.$$

We now note that $N_i \sim \text{Bin}(B, 1/d)$ and if $X \sim \text{Bin}(n, p)$, then $\mathbf{E}[X^2] = V(X) + \mathbf{E}[X]^2 = np(1-p) + n^2p^2$. Using this and denoting $B_d = \min\{B, d\} \leq d$, we get:

$$\mathbf{E}_{i_t^1, o_t^1, \ldots i_t^d, o_t^d}\left[\|\mathbf{g}_t\|_{\mathbf{w}_t}^*\right] \leq \frac{d}{2B_d\gamma_t}\sqrt{\sum_{i=1}^d \frac{3B_d}{d}} = \frac{d}{\gamma_t\sqrt{B_d}}.$$

$\qquad\square$

Finally we are now ready to proof the bound of our main theorem, Thm. 4:

*Proof of Thm. 4.* The proof follows almost the same steps that of proof of Thm. 1. In particular, same as the proof of Thm. 1, one can bound:

$$0.22\sum_{t=1}^T\left(\mathbf{u} - \mathbf{w}_t\right)^\top\boldsymbol{\theta}_t^* \leq \sum_{t=1}^T 2\eta\|\mathbf{g}_t\|_{\mathbf{w}_t}^{*2} + \frac{\nu\log\frac{1}{\delta}}{\eta}$$

$$\leq 2\eta\sum_{t=1}^T \frac{d^2}{\gamma_t^2 B_d} + \frac{\nu\log\frac{1}{\delta}}{\eta},$$

where the last inequality follows from Lem. 13. Same as before, choosing $\mathbf{u} := \arg\max_{\mathbf{x}\in\mathcal{D}_\delta}\sum_{t=1}^T\boldsymbol{\theta}_t^{*\top}\mathbf{x}$, and recalling that $\mathbf{x}^* := \arg\max_{\mathbf{x}\in\mathcal{D}}\sum_{t=1}^T\boldsymbol{\theta}_t^{*\top}\mathbf{x}$, we get:

$$\sum_{t=1}^{T} \left(\mathbf{x}^* - \mathbf{w}_t\right)^{\top} \boldsymbol{\theta}_t^* \leq \sum_{t=1}^{T} \left(u - \mathbf{w}_t\right)^{\top} \boldsymbol{\theta}_t^* + \delta TLD$$

$$= \frac{1}{0.22}\left[\eta \sum_{t=1}^{T} \frac{2d^2}{\gamma_t^2 B_d} + \frac{\nu \log \frac{1}{\delta}}{\eta}\right] + \delta TLD, \qquad \text{from Eq. (6)}$$

$$\leq \frac{1}{0.22}\left[\frac{\eta d^2 T}{\min\{1, H_{\mathcal{D},\psi}^2\}B_d} + \frac{\nu \log \frac{1}{\delta}}{\eta}\right] + \delta TLD, \quad \text{since we chose } \gamma \leq \min\{1, 0.7H_{\mathcal{D},\psi}\}$$

$$= \frac{d\sqrt{\nu T \log T}}{0.22 H_{\mathcal{D},\psi}\sqrt{B_d}} + LD,$$

choosing $\eta = \frac{\sqrt{\nu B_d} H_{\mathcal{D},\psi}}{d\sqrt{T \log T}}$ and $\delta = \frac{1}{T}$, concludes the proof noting the diameter of the decision set $\mathcal{D} \subseteq \mathcal{B}_d(1)$ is bounded by 1, and the lipschitz constant $L \leq \max_{t \in [T]} \|\boldsymbol{\theta}_t^*\| \leq 1$. $\qquad \square$

# D APPENDIX FOR SEC. 5

## D.1 *MNL-Scrible*: ALGORITHM PSEUDOCODE

---
**Algorithm 3 *MNL-Scrible***

---
1: **Input:** Initial point: $\mathbf{w}_1 \in \mathcal{D}$, Learning rate $\eta$, Perturbation parameter $\gamma$, Query budget $T$ (depends on error tolerance $\epsilon$), Batch-size $m$. Define $\ell_k := \lfloor \log m \rfloor$ and $\tilde{m} := 2^{\ell_k} \leq m$.
2: **Initialize** Current minimum $\mathbf{m}_1 = \mathbf{w}_1$
3: **for** $t = 1$ to $T$ **do**
4:     Compute: $\mathbf{w}_t = \arg\min_{\mathbf{w} \in \mathcal{D}_\delta} \left\{\eta \sum_{\tau=1}^{t-1}(-\mathbf{g}_\tau)^{\top}\mathbf{w} + \psi(\mathbf{w})\right\}$.
5:     Compute eigendecomposition s.t. $\nabla^2\psi(\mathbf{w}_t) = \sum_{i=1}^{d} \lambda_{t,i}\mathbf{v}_{t,i}\mathbf{v}_{t,i}^{\top}$.
6:     **for** $\ell = 1, 2, \ldots, m$ **do**
7:         Sample $i_t^\ell \in [d]$ uniformly at random.
8:         Choose $\mathbf{x}_t^\ell = \mathbf{w}_t + \gamma_t \frac{1}{2\sqrt{\lambda_{t,i_t^\ell}}}\mathbf{v}_{t,i_t^\ell}$ and $\mathbf{y}_t^\ell = \mathbf{w}_t - \gamma_t \frac{1}{2\sqrt{\lambda_{t,i_t^\ell}}}\mathbf{v}_{t,i_t^\ell}$.
9:         Play $(\mathbf{x}_t^\ell, \mathbf{y}_t^\ell)$, observe $o_t^\ell \sim \text{Ber}(P_t(\mathbf{x}_t^\ell, \mathbf{y}_t^\ell))$.
10:        $\mathbf{g}_t^\ell = \frac{d}{\gamma_t}\left(o_t^\ell - \frac{1}{2}\right)\sqrt{\lambda_{t,i_t^\ell}}\mathbf{v}_{t,i_t^\ell}$.
11:     **end for**
12:     Update $\mathbf{g}_t = \frac{1}{m}\sum_{\ell=1}^{m}\mathbf{g}_t^\ell$
13: **end for**
14: **for** $t = 1, 2, 3, \ldots, T$ **do**
15:     Sample $\mathbf{u}_t^1, \mathbf{u}_t^2, \ldots \mathbf{u}_t^{\ell_k} \overset{\text{iid}}{\sim} \text{Unif}(\mathcal{S}_d(\frac{1}{\sqrt{\ell_k}}))$. Denote $U_t := [\mathbf{u}_t^1, \ldots, \mathbf{u}_t^{\ell_k}] \in \mathbb{R}^{d \times \ell_k}$
16:     Define $S_t := \{\mathbf{w}_t + \gamma U_t\mathbf{v} \mid \mathbf{v} \in V_{\ell_k}\}$ (see definition of $V_{\ell_k}$ in the description)
17:     Play the $m$-subset $S_t$
18:     Receive the winner feedback $o_t = \arg\min(f(\mathbf{x}_t^1), f(\mathbf{x}_t^2), \ldots, f(\mathbf{x}_t^{\tilde{m}}))$
19: **end for**
20: Return $\mathbf{m}_{T+1}$

---

**Lemma 14** (Pairwise Properties of MNL Model).

$$P(i > j) = \sum_{\sigma \in \Sigma_{i,j}} P(\sigma)$$

# E SOME USEFUL RESULTS

**Lemma 15.** *For any $x \in [-0.7, 0.7]$, $\sigma'(x) \in [0.222, 0.25]$.*

*Proof.* Let us first consider the positive interval $x \in [0, 0.7]$. Note by definition, since $\sigma(x) = \frac{1}{1+e^{-x}}$, $\forall x \in \mathbb{R}$, first derivative and the second derivative of sigmoid is respectively given by:

$$\sigma'(x) = \frac{e^{-x}}{(1 + e^{-x})(1 + e^{-x})}$$

$$= \frac{1}{1 + e^{-x}} - \frac{1}{(1 + e^{-x})^2},$$

and

$$\sigma''(x) = \left[\frac{e^{-x}}{(1 + e^{-x})^2} - \frac{2e^{-x}}{(1 + e^{-x})^3}\right].$$

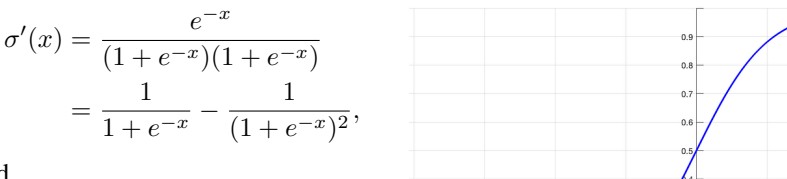

As shown in the right figure, this brings us to the observation that $\sigma''(x) < 0$ for any $x > 0$, and thus $\sigma'(\cdot)$ is a decreasing function in the interval $[0, \infty)$. Thus the function $\sigma'(\cdot)$ attains maximum at $x = 0$ and minimum at $x = 0.7$, yielding $\sigma'(x) \in [0.222, 0.25]$ in the range $x \in [0, 0.7]$.

The result follows from the symmetry of $\sigma'(\cdot)$ function around the $Y$-axis. $\qquad \square$