# OpenReview forum: "Dueling in the Dark: An Efficient and Optimal $O(\sqrt{T})$ Mirror Descent Approach for Competing against Adversarial Preferences"
_ICLR.cc/2025/Conference — Submitted to ICLR 2025_

### Official Review · Reviewer_QdtU · 2024-10-20

**Soundness:** 3
**Presentation:** 3
**Contribution:** 3
**Rating:** 5
**Confidence:** 4

**Summary:**

This work studies the linear dueling bandit problem with adversarial binary feedback. The author proposes a novel algorithm based on the Mirror Descent method and achieves a near-optimal regret guarantee. In addition, the author extends the algorithm to deal with batched datasets or multinomial bandits, maintaining a similar regret guarantee.

**Strengths:**

1. The author proposes a novel algorithm based on the mirror descent method and achieves a near-optimal regret guarantee.

2. The author generalizes the result to several bandit settings with batched datasets and multinomial logit bandits.

3. The paper is well-written and easy to follow.

**Weaknesses:**

My main concern is that the author may be overclaiming some contributions.

1. As the author mentioned when discussing the limitations of existing algorithms, prior attempts to adapt them to RLHF either make unrealistic modeling assumptions or are computationally inefficient. The main contribution of this work is claimed to be the development of more computationally efficient algorithms. However, this work focuses only on the logistic linear bandit, which also falls under the category of unrealistic modeling assumptions and does not offer a clear advantage over previous algorithms for logistic linear bandits. Additionally, under this setting, an efficient algorithm [1] already exists, even with adversarial preferences.

[1] Nearly Optimal Algorithms for Contextual Dueling Bandits from Adversarial Feedback

2. The author claims a contribution for providing a lower bound on the regret when learning a logistic dueling bandit. However, a lower regret bound already exists in previous work [2].

[2] Stochastic Contextual Dueling Bandits Under Linear Stochastic Transitivity Models

3. The author mentions that gradient descent algorithms are simple to implement and can seamlessly integrate with modern deep learning frameworks, making these methods computationally efficient. However, the proposed method only focuses on the logistic dueling bandit, and it is unclear how to implement it in modern deep learning frameworks or whether the performance guarantees still hold in such settings. Furthermore, when restricted to the logistic dueling bandit problem, the proposed method does not seem to offer an advantage over UCB or Thompson Sampling (TS) methods.

**Questions:**

See Weaknesses.

---

> ### Author Response · Authors · 2024-12-02
> **Rebuttal for  Reviewer QdtU**
>
> Thanks for your valuable feedback. We clarify your specific questions below:
>
> > W1: My main concern is that the author may be overclaiming some contributions. Prior attempts to adapt them to RLHF either make unrealistic modeling assumptions or are computationally inefficient. The main contribution of this work is claimed to be the development of more computationally efficient algorithms. However, this work focuses only on the logistic linear bandit, which also falls under the category of unrealistic modeling assumptions and does not offer a clear advantage over previous algorithms for logistic linear bandits. Additionally, under this setting, an efficient algorithm [1] already exists, even with adversarial preferences.
>
> -- We appreciate your feedback, and thanks for raising the concern and allowing us to clarify: With all respect, we however firmly disagree with the comments and implications pointed in W1. Note the main focus of this work is designing computationally efficient algorithms for RLHF problems with theoretically sound performance guarantees. We clarify each points one by one below:
>
> -- *Re. unrealistic modeling:* We assume continuous decision space and adversarial preference model, which itself is general enough and something that is not considered in many prior works. For continuous decision spaces, the most general preference model that has been studied so far is based on a sigmoid of score differences where the probability of item ${\mathbf x}$ being preferred over ${\mathbf y}$ is given by
> $$
> P({\mathbf x} \succ {\mathbf y}) = \sigma(f({\mathbf x})-f({\mathbf y})),
> $$
> $f: {\mathcal D} \mapsto {\mathbb R}$ being a general scoring function. Usually $f$ is considered to be linear function as also considered in most of the prior works in RLHF literature (Saha'21, Bengs et al'22, Pacchiano et al'23, Kausik et al'24, Wang et al'23, Das et al'24, Di et al.'24, Li etal'24, Wu \& Sun'24). We already explained in Q1 of Reviewer iPuV how our preference model and theoretical guarantees could be extended to the above preference model. Is the reviewer familiar with any other more general preference modeling for continuous action spaces? Of course, one can model the preference function to be as general as a deep neural network, but given the limited theoretical understanding of basic learning theory guarantees of neural nets, it is obviously beyond the scope of this paper to provide online theoretical guarantees on neural net-based preference models. Hence the claim of the reviewer about ``unrealistic modeling assumptions" stands vacuous.
>
> -- *Re. ``does not offer a clear advantage over previous algorithms, e.g. in [1]":* Thanks for pointing to [1], which is certainly a related work and will cite in the update. However, as mentioned repeatedly in our main paper as well as in the rebuttal, the main disadvantage of UCB-based methods lies in the computation complexity of the optimism-based update step, e.g. **it is not explained in [1] how do you compute the pair $(a_t,b_t)$ in Line 6 of Alg-1 (RCDB)** when the decision space ${\mathcal A}$ is infinite and continuous. The optimization function is not convex in $a_t, b_t$, so how do you optimize that step? --- **It is computationally infeasible**! The authors of [1] also hence chose to work with a really simple dataset with $d =2$ and a finite action space ${\mathcal A}$ of just $32$ arms! In fact, if you look at the rest of the UCB-based algorithms, e.g., Bengs et al.'22 or Saha'21, they all used finite arms (~$50$ arms in both papers) and very small dimension $d = 10$, precisely since their algorithms do not scale, and the optimization routine is computationally infeasible for continuous decision spaces. Our method is the first method that can elegantly handle any continuous decision space $\subset {\mathbb R}^d$, as we showed in our experiments section. In fact, not only do we work with continuous decision spaces with $O(\sqrt T)$ guarantee, but our algorithm elegantly performs with large dimensions within a matter of few seconds! (Please check our reported experiments in the updated draft). Please also see our response in **W3** below for a related discussion.

---

> > ### Author Response · Authors · 2024-12-02
> > **Contd: Rebuttal for Reviewer QdtU**
> >
> > -- *Re. ``[1] also works in adversarial setting":* Firstly, (i). please note this is an **extremely misleading statement** since the paper [1] uses the word `adversarial' to model a different setting of robustness or corrupted feedback. This is very different from the adversarial preference (modeled by ${\boldsymbol \theta^*}_1,\ldots,{\boldsymbol \theta^*}_T$) we considered in our paper. This is evident simply from the preference modeling of [1] that assumed a fixed scoring function $r^*$ modeled with a fixed unknown parameter ${\boldsymbol \theta^*}_t$ (see Eqn. 3.1 and Assumption 3.1 of [1]).
> > Secondly (ii). even if we assume the corruption model of [1] is equivalent to our adversarial model $({\boldsymbol \theta^*}_1,\ldots,{\boldsymbol \theta^*}_T)$, it only reflects the superiority of our method and theoretical guarantees as our preference model parameters ${\boldsymbol \theta^*}_1,\ldots,{\boldsymbol \theta^*}_T$ are completely arbitrary, the number of corruption rounds (a.k.a. the total number of adversarial feedback) $C$ in [1] could very well be $O(T)$ in our setting. But note [1]'s regret guarantee is $O(d\sqrt{T} + dC)$, which boils down to a trivial $O(dT)$ regret bound for $C = O(T)$! While our results still guarantee a $O(d\sqrt{T})$ bound even if $C = O(T)$ which immediately sets a clear advantage of our proposed methods and regret analysis over [1]. Thus, the claim that [1] works in an adversarial setting also stands vacuous!
> >
> > -- *Re. Additional advantages of our work over [1]:* Finally, since we are comparing our work with [1], we can in fact point the reviewers to several other advantages/improvements of our work over [1], which is worth noting:
> > - We considered the $B$-batched preference feedback setting (Sec 4), which is more realistic from a practical point of view as online/active updation of the learning algorithm after each feedback is costly (and perhaps impractical for distributed learning environments). See Thm 4, Cor 5.
> > - We also considered a generalization of dueling feedback $(k=2)$ for $k$-sized subsetwise feedback with top-$m$ rank-ordered preferences. See Thm 7 and Cor 8.
> > - We provided the first lower bound for the $B$-batched and top-$m$ rank-ordered preference feedback settings. See Thm 6 and Thm 9.
> > - We also experimentally validated our proposed algorithm for high dimensions \red{$d = 50$} and for continuous decision spaces. Please read our response above, where we explained how the contemporary methods (Di et al.'24, Bengs et al.'22, Saha'21) fail to run their experiments beyond a finite decision space ${\mathcal D}$ with small number of arms.
> >
> > > W2: Contribution in Lower Bound over [2] ([2] Stochastic Contextual Dueling Bandits Under Linear Stochastic Transitivity Models)
> >
> > -- Thanks for the pointer, but we are already quite familiar with this work, as we have also cited this in our paper. Please note, this work is first a purely dueling bandit work, which does not consider any general form of feedback, e.g.\ batched or top-$m$ ranking feedback. We did not claim our contribution lies in deriving lower bounds in the dueling (pairwise preference) setting. Our novelty in the lower bound analysis lies in proving the improved lower bounds for these multiway preference feedback models, which is first in this line of work (see Thm 4 and 6).

---

> > > ### Author Response · Authors · 2024-12-02
> > > **Contd: Rebuttal for Reviewer QdtU**
> > >
> > > > W3: The proposed method only focuses on the logistic dueling bandit, and it is unclear how to implement it in modern deep learning frameworks or whether the performance guarantees still hold in such settings.
> > >
> > > We appreciate the reviewer’s concern about the implementation of the proposed method in modern deep-learning frameworks and the associated performance guarantees. However, note, as we also mentioned in W1 above, there is limited foundational work in deep learning theory, particularly in deriving learning-theoretic guarantees for deep learning algorithms, which is a first step before deriving any online DL and RLHF-DL theoretic guarantees. Addressing such theoretical challenges would require advancing the understanding of basic learning guarantees for deep learning models before attempting to establish guarantees for RLHF, which inherently adds further complexity. Thus, while extending our theoretical guarantees to deep learning-based preference models is an exciting direction for future research, it is naturally far beyond the scope of this paper. However, since deep learning algorithms are primarily based on gradient descent techniques (it is easy to compute gradients for neural nets), our gradient-based routines easily integrate with any deep-net model, modulo any theoretical guarantees. Hence we believe our methods lay the essential groundwork that can later inform studies involving deep learning-based preference models as the learning theory literature makes progress in the fundamental understanding of deep learning theory.
> > >
> > > We also urge the reviewer to kindly refer to our common response. We added experimental evaluation in the updated draft (in the Openreview portal) and reported the regret performance with different types of feedback models studied in the paper, as well as the runtime performances. Our experimental results provide evidence of the runtime efficiency of our methods even for dimension $(d)$ as large as $d=50$ and continuous decision spaces, while the prior works only reported their experiments for finite arms (Di et al' 2024 32 arms, Saha 2021 and Bengs et al' 2022 50 arms, etc), and really small dimension $d$ (Di et al' 2024 considered $d=2$, Saha 2021 and Bengs et al' 2022 used $d=10$, etc).
> > >
> > > > W4: The proposed method does not seem to offer an advantage over UCB or Thompson Sampling (TS) methods.
> > >
> > > -- Again, we respectfully but firmly disagree. We mentioned the limitations of UCB and TS-based works in Line 082-084 --- the line of work based on optimism in the face of the uncertainty (UCB) principle, which requires maintaining confidence sets and optimizing over continuous policy/action space, which makes their algorithm computationally intractable. Note we explained in W1 above how the UCB-based algorithms (Di et al.'24, Bengs et al.'22, Saha'21) actually reported their experiment results on a small finite action space since their arm-selection rules are computationally infeasible for continuous decision spaces. But we successfully reported experiments in continuous decision spaces unlike the prior UCB works. Similarly, for TS works, updating the posterior distribution becomes computationally inefficient beyond normal distributions, especially for large action spaces. e.g., in Alg 1 (FGTS.CDB) of Li et al.'24, the closed form of the posterior update $p^j(\cdot)$ is computationally infeasible, and consequently, the sampling step of $\theta_t^j \sim p^j(\cdot)$ is also infeasible. Similar problems occur in Alg 2 (PbTS) of Wu \& Sun'24 (see Line 3 and 4 of Alg 2). This is also the reason why we could not implement these UCB / TS-based algorithms in our empirical evaluation, which does not scale for continuous decision spaces. Further their UCB and TS methods do not apply to adversarial functions as well, unlike our proposed methods. Neither of these existing methods applies to more general preference models, e.g., batched preferences or top-$m$ ranking feedback.
> > >
> > > ---
> > >
> > > Please let us know if you have any remaining questions, would be happy to clarify. We would greatly appreciate it if you could reconsider the scores based on the above clarifications.

---

> > > > ### Comment · Reviewer_QdtU · 2024-12-03
> > > >
> > > > Thank you for the authors’ feedback. I have decided to keep my score unchanged. Below are my comments:
> > > >
> > > > 1. Regarding the preference model for human feedback, exploring a general score function is an interesting direction. However, this aspect is not included in the paper, making it difficult to verify the correctness of the claim without a detailed proof. Additionally, a more general case is presented in the total general preference framework [1], where the transitivity property may not hold. It is straightforward to confirm that for any score function, the BT model always satisfies the transitivity property.
> > > >
> > > > [1] A Minimaximalist Approach to Reinforcement Learning from Human Feedback
> > > >
> > > > 2. Based on the feedback provided, it seems the main advantage of this method lies in its application to continuous decision spaces, a point that was not explicitly stated in the paper.

---

> ### Author Response · Authors · 2024-12-03
> **Further clarifications to Reviewer QdtU**
>
> Dear Reviewer QdtU,
>
> Thank you for your feedback. We appreciate you pointing us to different RLHF papers for comparison. We want to assure you that we have thoroughly studied the RLHF theory literature before writing this paper and have been keeping a close eye on recent developments ever since. Additionally, we want to emphasize that we strive to avoid publishing incremental or marginal contributions (as opposed to many existing RLHF papers). We spent years on this work before deciding to submit it, ensuring that it has sufficient "meat" and clearly stands out among the existing learning-with-preferences and RLHF works. Publishing superficial or incremental contributions is not something we also consider worth our time and effort.
>
> Thus, while we appreciate you pointing us to more papers in the RLHF theory literature (and certainly cite them in the update), we are confident in our ability to rebut comparisons and demonstrate why our results stand out from those works.  We would also like to confirm that the paper you pointed out, [1], is one we are closely familiar with, and we are well aware of its comparative limitations.  Let us point them out one by one:
>
> **Re. Comparison to [1] (Swamy et al., 2024)**  *The two key limitations* of this work lie in their • ((i)) performance bounds (as provided in [1]) are really weak, especially for large/ general continuous decision spaces. This is due to their general preference modeling assumptions, which prevent them from exploiting the additional structure of the model, making the theoretical bound vacuous for real-world scenarios. The second major problem again lies in their • ((ii)) algorithmic inefficiency, where they fail to propose an algorithm that simultaneously respects their theoretical guarantees (despite them being weak) as well as runtime efficiency. Due to this, they had to propose two different algorithms separately - one for theory (SPO, Alg1) and one for practice (SPO, Alg 2).  Below, we elaborate these in more detail:
>
> 1. **Performance Guarantees**:
>    - The theoretical guarantees in [1] are poor. For example, Theorem 2.3 does not provide a regret guarantee; it states that they can find an $\frac{2Reg(T)}{T}$-approximate Minimax Winner. However, no explicit bounds are given for $Reg(T)$, apart from the fact that it is sublinear. This leaves room for $Reg(T)$ to be as large as $O(T^{0.9999})$, which is significant.
>    - The second performance bound, Corollary 2.5 (and Corollary 2.6 derived from it), states that if the best decision (policy) is well-separated, the solution achieves an $\tilde O\left(\frac{| \Pi |}{\Delta T}\right)$-approximate Minimax Winner guarantee (or $O\left(H\sqrt{\frac{\log |A|}{T}}\right)$-approximate Minimax Winner for Corollary 2.6). However, this guarantee is vacuous when the policy space $|\Pi|$ or action space $|A|$ (equivalent to our decision space $\mathcal{D}$) is infinite, continuous, or even very large.
>
> 2. **General Preference Models Comes at the Cost of Poor Theoretical Guarantees for General Decision Spaces**:
>    - While [1] adopts a general preference model without transitivity assumptions, this comes at a significant cost of poor regret guarantees, especially in large or continuous decision spaces.
>    - General preference models fail to exploit the low-dimensional structure (e.g., arm similarities), leading to weak performance guarantees beyond finite decision or tabular RL settings.
>
> 3. **Sub-Optimality of approximate Minimax Winner guarantee of [1] is different and weaker than our Regret guarantees**:
>    - Unlike our work, [1] does not provide regret guarantees. Their bounds are only on the suboptimality of the "approximate Minimax Winner," which differs significantly in nature from our regret guarantees.
>    - This is also why [1] does not report regret performance in their experiments, further highlighting the disparity between the guarantees provided by our work and theirs.
>
> 4. **Computational Inefficiency of their proposed methods**:
>    - [1] introduces two algorithms: Algorithm 1 (SPO, the theoretical version), for which the above guarantees hold. However, it is computationally inefficient in multiple steps (e.g., Lines 6, 7, and 11).
>    - To address this, [1] proposes Algorithm 2, a practical version of SPO, which unfortunately lacks any theoretical guarantees.
>    - Therefore, it is **unfair to compare** our algorithms to those in [1], as our methods are both computationally efficient and provably regret-optimal.
>
> In summary, [1] cannot effectively handle continuous or infinite (or even very large) decision spaces, as their theoretical guarantees break down in these scenarios. They do not provide regret guarantees (even their practical SPO algorithm requires querying a batch of size $B$, leading to poor regret bounds). Meanwhile, our work delivers regret guarantees for various feedback settings and simultaneously offers runtime-efficient and theoretically sound algorithms.

---

> ### Author Response · Authors · 2024-12-03
> **Contd: Further clarifications to Reviewer QdtU**
>
> **Re. "exploring a general score function is an interesting direction. However, this aspect is not included in the paper, making it difficult to verify the correctness of the claim without a detailed proof"**
>
> We agree that exploring general score functions is an interesting direction. However, as mentioned, this is a very broad question and lies outside the scope of this work. Our current work represents a first step toward efficient RLHF algorithms for continuous decision spaces under linear score functions. Extending this to convex or more general scoring functions is indeed an interesting open problem, but addressing it would require significant non-trivial effort in terms of algorithm design and theoretical analysis. Hence it seems unfair to mention/consider not addressing such a broad general complex direction as a limitation to this work. This was not even the focus of this work, but certainly interesting for future directions. In fact, multiple directions and papers could be produced from such an open-ended general broad direction with different types of feedback models, different types of general scoring/preferences functions, different algorithmic approaches and efficiency, etc.
>
> ---
>
> **Re. "...Based on the feedback provided, it seems the main advantage of this method lies in its application to continuous decision spaces, a point that was not explicitly stated in the paper."**
>
> We are glad that you acknowledge this important contribution of our work. However, we would like to emphasize that this is only one of several contributions. To briefly summarize, our work addresses:
> - General continuous (potentially infinite) decision spaces.
> - Multiple feedback models.
> - Adversarial models.
> - Regret guarantees that are both provably optimal
> - Our theoretical results are supported by strong empirical evidence.
>
> (See our list of contributions in Section 1 for more details.)
>
> *We would also like to point out that this contribution has been mentioned multiple times in the paper*. For example:
> - In the abstract: *"...are limited to small-scale problems due to computationally intractable steps. We address this challenge..."*
> - Line 183 specifies a general decision space, \(\mathcal{D} \subseteq \mathbb{R}^d\).
>
> *However, we appreciate your feedback and will make this contribution more explicit throughout the paper and in the related works section.* Thank you.
>
> ---
>
> Given the above clarifications, we earnestly request you to kindly reconsider your score. We are confident about the novelty and uniqueness of our results compared to existing RLHF theory work.
>
> Sincerely,
> The Authors

---

### Official Review · Reviewer_4UGY · 2024-10-29

**Soundness:** 3
**Presentation:** 3
**Contribution:** 3
**Rating:** 6
**Confidence:** 2

**Summary:**

This paper presents a mirror descent approach for a variety of online linear optimization (OLO) problems that involve preference feedback.  The work presents matching upper- and lower-regret bounds (up to logarithmic factors) for each of the scenarios considered.  First, the authors consider adversarial logistic dueling bandits and present the Double-Scrible algorithm with matching $O(\sqrt{T})$ upper and lower regret bounds. The authors generalize this work to the batched setting and present the BaBle-Scrible algorithm we equivalent matching regret bounds.  Finally, the authors show that the top-m ranking feedback setting can be reduced to the batch setting and they present MNL-Scrible with equivalent matching regret bounds.

**Strengths:**

This is a reasonably well written paper that is easy to read despite its highly technical nature.  The work addresses a seemingly important problem of OLO with preference feedback.  By the author's admission this is the first efficient online gradient descent-based algorithm for this problem with provably optimal performance guarantees.  The paper does a good job of considering generalizations of the original dueling bandits framework by incorporating batched responses and ranked preferences.

**Weaknesses:**

**UPDATE** The AC should be aware that the current draft of this paper contains a significant amount of content on page 11.  It thus risks a desk reject.  I am comfortable with the initial version of the paper, without experiments, which fits in the 10-page limit.

**NOTE** I do not consider myself qualified to adequately review this paper as I am not an ML theorist.  I do not have sufficient knowledge of the literature in this space to judge impact / contribution.  I have notified the AC to this effect early in the review process but did not receive a response.  Nevertheless, below is my best attempt at a critique for this work and I will assign a low confidence score to help calibrate.

Rather than do one thing well this paper reaches for a lot of potential contributions that in my opinion lessen the overall impact of the work.  In particular, the references to LLMs in the abstract and introduction seem a bit misplaced as it is unclear what the actual connection is to LLMs in this work (marketing?).  The authors present three algorithms with regret bound analysis, but do not provide any guidance on implementation or empirical validation, making it impossible to conclude whether the algorithms are effective in practical settings.

One of my issues in reviewing this work is that I cannot place it in the context of existing work.  The paper would benefit from a "Related Work" section to help highlight the potential impact of these contributions w.r.t. existing work.

As a broader comment, ICLR is a conference on "learning representations" and it is unclear what connection this work has to representation learning.  At first glance it would not seem that this work is well-placed at ICLR.  Upon further digging it seems that ICLR has published quite a few works on bandit algorithms and so perhaps the organizers better insight into the fitment of this work to the conference.

**Questions:**

See weaknesses.

---

> ### Author Response · Authors · 2024-12-02
> **Rebuttal for Reviewer 4UGY**
>
> Thanks for your valuable feedback. We clarify your specific questions below:
>
> **W1.** Connection to LLMs
>
> We would like to address your concerns by first providing an overview of the literature on theoretical RLHF which all claims applications to LLM fine-tuning through RLHF. E.g., Liu et al'24, Saha etal'23, Kausik etal'24, Wang etal'23, Das etal'24, Di etal'24, Li etal'24, Wu \& Sun'24, Mukherjee etal'24 are among the many works that focus on theory of RLHF and address the same general problem setting, where the goal is to "learn" from human preferences. These works commonly adopt models such as the sigmoid-based or BTL model, as described in our problem setup (Section 2.1). **Notably** they also highlight LLM fine-tuning as part of their primary motivations.
>
> Our work further improves these prior contributions in several ways:
>
> 1. Novel Algorithmic Approach
> - We propose a unique approach to the problem using OMD-based algorithms, which adopt a gradient descent flavor. This stands in contrast to previous works that predominantly rely on UCB or TS techniques.
> - As a result, our algorithms provide not only computational advantages but also achieve regret-optimal performance, as demonstrated in Thm1,3 & Cor2. Please also refer to **W1(II)** for R-iPuV for more details of our key algorithmic novelty.
>
> 2. *Multiset Feedback*
> - Unlike existing studies, we extend our results to multiset feedback scenarios, including *B-batched feedback* (Thm4,6, & Cor5) and *top-m ranking feedback* (Thm7,9, & Cor8). These extensions result in improved performance guarantees, offering broader applicability of our methods.
>
> 3. *Adversarial Models*
> - While prior works assume a stochastic RLHF environment, we generalize the problem to a dynamic adversarial feedback model.
> - This modeling makes our approach robust and well-suited for dynamically evolving preferences, which are frequently encountered in LLM fine-tuning settings.
> - We urge the reviewer to consider classical literature in adversarial online learning and adversarial bandit theory (e.g., Auer et al,2002; Zinkevich,2003; *Introduction to Online Convex Optimization* by Hazan; and *Bandit Algorithms* by Lattimore & Szepesvári).
> - The adversarial model inherently accounts for noise due to human feedback, dynamically changing environments, and robustness in learning algorithms. Moreover, as adversarial models generalize stochastic models, our problem setup and proposed algorithms retain their performance guarantees under stochastic preference feedback models as is.
>
> In summary, our framework generalizes prior RLHF work cited above and directly applies to LLM fine-tuning scenarios, as also motivated by these prior works.
>
> To further understand the connection to LLM fine-tuning as an RLHF problem at an intuitive level, note after learning a bse LLM through supervised learning (preprocessing step in training LLMs), the subsequent step of LLM fine-tuning heavily relies on human preferences or ranking feedback. In LLM applications, the decision space $\mathcal{D}$ corresponds to the set of all potential answers to a given question. The goal is to identify the best answer as quickly as possible using sequential human feedback. In this context, our method addresses the problem of learning the best response to a prompt (the "best decision" in decision space $\mathcal{D}$) through regret minimization or best-arm identification objectives (in Sec 2.1, 2.2, 2.3) making it well-aligned with the requirements of LLM fine-tuning.
>
> **W2.** Empirical evaluation: Thanks for the suggestion. Please refer to our common response. We added experimental evaluation in the updated draft (in the Openreview portal) and reported the regret performance with different types of feedback models studied in the paper, as well as the runtime performances.
>
> **Re. "Rather than do one thing well this paper reaches for a lot of potential contributions"** We appreciate your feedback. Pls note we presented 3 algorithms as part of our theoretical contribution, as common in theory focused works. Since the problems addressed in Alg2 and 3 is related to (in a way a generalization of) that of Alg1, we believe it is best to add these algorithms in the same paper, rather than publishing multiple papers on similar/ related questions. Therefore, we believe these additional results would strengthen our work, *we also added empirical results for these settings in the updated draft*.
>
> **Re. Related Works section** (for defining the context of this work): Please note we included a discussion of the related work in the section titled "Limitations of Existing Online RLHF with Preference Feedback Algorithms''.  To make it clear we would be happy to rename that section to "Related Works'' and also include any additional reference you may suggest.
>
> ---
>
> Please let us know if you have any remaining questions, would be happy to clarify. We would greatly appreciate it if you could reconsider the scores based on the above clarifications.

---

> > ### Author Response · Authors · 2024-12-03
> > **Follow-up**
> >
> > Dear Reviewer 4UGY,
> >
> > Thank you once again for taking the time to review our paper and provide thoughtful and valuable feedback. We hope that our rebuttal has addressed your concerns.
> >
> > As the discussion period approaches its conclusion, we kindly ask if we could address any additional questions or comments that we could clarify in the remaining time.
> >
> > If you think that our responses have sufficiently addressed your concerns, we would be truly grateful if you could reconsider the evaluation of our paper based on the clarification provided in the rebuttal.
> >
> > Thanks, Authors

---

### Official Review · Reviewer_rg2J · 2024-10-30

**Soundness:** 3
**Presentation:** 3
**Contribution:** 3
**Rating:** 6
**Confidence:** 4

**Summary:**

This paper focuses on the online convex optimization problem from preference and ranking feedbacks, which is a simplification of the modern RLHF framework. The paper studies the adversarial online linear optimization problem where the utility of each arm is linear. At each round, the agent plays two arms and the feedback on which arm is better is generated based on the Bradley-Terry model. The goal of the agent is to minimize the utility regret. The paper proposes an online mirror descent algorithm named Double-Scrible for the pair-wise preference setting, with provable regret guarantee matching the lower bound. Generation to batched settings and ranking feedbacks are also studied with theoretically optimal algorithms.

**Strengths:**

1. Novelty and Impact: This paper studies the online convex optimization problem with preference feedback, the problem is of importance and directly related to solving RLHF problems in practice. The insight of the proposed algorithm has potential in many RLHF application areas.

2. Theoretical Soundness: For all three feedback settings, i.e., pair-wise preference feedback, batched feedback, and ranking feedback settings, the paper propose gradient based online optimization algorithms which has theoretical regret guarantees matching the lower bound. Overall it is a solid theoretical paper with good contribution.

3. Algorithm Practicality: the proposed algorithm is computationally tractable with gradient based approach, the major computation burden in each iteration seems to come from eigen decomposition.

**Weaknesses:**

1. Motivation: the motivation of study adversarial human feedback (the linear parameter \theta_t adversarially change over time) seems weak from the presentation of paper.

2. Comparison: the regret definitions are different for each feedback settings, which makes comparing the bounds across settings unfair. For example, the bound in Theorem 3 accounts for the case where the sample size and the number of human queries are B times larger than the case in Theorem 1 (although the definition of regret in batched setting is averaged). Therefore, the claim that batched comparison and ranking feedback improves algorithm performance is not completely solid, although correct.

3. Computation Efficiency: one of this paper's claim is the proposed algorithm is more computationally efficient compared to confidence based algorithms (UCB or TS based) in the literature, but the evidence is not clear from the paper. Given computation tractability is a major strength of the proposed algorithm, the reviewer feels that numerical evidence should be provided, at least for a simple setup.

Typos:

1. equation 1 seems to be a typo

2. Theorem 3, superscript of regret should be Batched-LogitDB

3. line 458, missing reference.

**Questions:**

1. Can you provide more examples on why adversarial human preference is of importance in RLHF applications, to motivate your theoretical framework?

2. Can you provide numerical evidence showing the computation efficiency of your proposed algorithm?

3. Can you present the regret bound of the three settings in a way that can be fairly compared?

4. In line 347, you claim your runtime requirement is O(dT), it seems does not count the computation complexity of eigen decomposition?

---

> ### Author Response · Authors · 2024-12-02
> **Rebuttal for Reviewer rg2J**
>
> Thanks for your valuable feedback. We clarify your specific questions below:
>
> > W1: Motivation for studying adversarial feedback in the context of RLHF?
>
> - Thanks for the great question; however, we beg to differ from the statement that "The motivation for studying adversarial human feedback (parametrized through $\theta_t$) is weak" since
> adversarial models are only more general than stochastic preference models (i.e. stochastic models are only a special case of adversarial models). So our problem setup and proposed algorithms retain their performance guarantees under stochastic preference feedback models as is.
>
> **Re. the connection to RLHF** We would like to address your concerns by first providing an overview of the literature on theoretical RLHF: E.g., Liu et al'24, Saha etal'23, Kausik etal'24, Wang etal'23, Das etal'24, Di etal'24, Li etal'24, Wu \& Sun'24, Mukherjee etal'24 are among the many works that focus on theory of RLHF and address the same general problem setting, where the goal is to "learn" from human preferences. These works commonly adopt models such as the sigmoid-based or BTL model, as described in our problem setup (Section 2.1). **Notably** they all highlight RLHF problem setup as part of their primary motivations.
>
> In slightly more detail, note RLHF studies the learning with human feedback problem in general, and human-preferences are considered to be one of the primary forms of human expression (feedback). And our work focuses on the same problem framework quantified in terms of regret minimization or best arm identification objective (as defined in our problem setup, Sec 2.1,2.2,2.3). Notably, our objectives are designed to converge to a "good decision."
>
> Suppose we consider our decision space $(\mathcal{D})$ as a set of trajectories (or distributions over trajectories), which is a common scenario in general RLHF applications. Examples include:
>  *(i) Robotics*, where trajectories represent the actions taken by the robot, *(ii) LLM fine-tunings*, where trajectories correspond to the answers generated by the LLM in response to a prompt,
> *(iii) Autonomous vehicles*, where trajectories define the path taken by the vehicle, *(iv) Recommender Systems*, where trajectories could be the sequence of items recommender to the customer, etc.
>
> Now based on the objectives defined in Sec 2, learning the best decision (i.e., trajectory) in our setting implies that the algorithm learns to identify the desired action through sequential human feedback (preferences). Our optimal regret (Thm1,4,7) and sample complexity guarantees (Cor2,5,8) justify the desired performance of our algorithms through matching lower bound guarantees (Thm 3,6,9). Consequently, our approach is applicable to a wide range of RLHF problems, justifying our motivation for RLHF applications.
>
> **Re. Adversarial Models** Please note while prior works (cited above) did address a similar problem framework as ours, they predominantly considered a stochastic RLHF environment, while we generalize the problem to a dynamic adversarial feedback model.
> - This modeling makes our approach robust and well-suited for dynamically evolving preferences, which are frequently encountered in LLM fine-tuning settings. This is more practical for modeling real-world human feedback to incorporate the noise, fatigue, inattentiveness, shift of preferences in real human feedback, etc.
> - We also urge the reviewer to kindly go over the classical literature in adversarial online learning and adversarial bandit theory (e.g., Auer et al,2002; Zinkevich,2003; *Introduction to Online Convex Optimization* by Hazan; and *Bandit Algorithms* by Lattimore & Szepesvári) which is a well-studied literature of online learning theory developed primarily with the motivation to incorporate the above mentioned dynamic aspects of feedback models that stochastic models fails to capture.
> - The adversarial model also accounts for the robustness of the resulting learning algorithms when exposed to adversarial perturbations due to noise, data corruption, breaching attacks, etc.
>
> Hope that clarifies your question regarding the **connection to RLHF and motivation behind the adversarial preference models**. It is also worth pointing out some of the other advantages of our work compared to the prior works (other than (#1) *Generalizing the results to adversarial RLHF feedback models*):
>
> #2. *Novel Algorithmic Approach*
> - We propose a unique approach to the problem using OMD-based algorithms, which adopt a gradient descent flavor. This stands in contrast to previous works that predominantly rely on UCB or TS techniques.
> - As a result, our algorithms provide not only computational advantages but also achieve regret-optimal performance, as demonstrated in Thm1,3 & Cor2. Please also refer to **W1(II)** for R-iPuV for more details of our key algorithmic novelty.

---

> > ### Author Response · Authors · 2024-12-02
> > **Contd: Rebuttal for Reviewer rg2J**
> >
> > #3. *Multiset Feedback*
> > - Unlike existing studies, we extend our results to multiset feedback scenarios, including *B-batched feedback* (Thm4,6, & Cor5) and *top-m ranking feedback* (Thm7,9, & Cor8). These extensions result in improved performance guarantees, offering broader applicability of our methods.
> >
> > > W2 (Q3): The regret bounds of the three settings are defined differently, making comparisons unfair. Can the authors present the regret bounds in a way that can be fairly compared?
> >
> > - Thanks for the question. We are, however, slightly worried that there is probably a misunderstanding since our regret bounds are unified across all three settings (Logit-DB, $B$-Batched Logit-DB and MNL-bandits with top-$m$ ranking feedback) in order to make the comparisons fair. On a high level, at each round we penalize each action $w_t$ against the best item $x^\star$ in terms of their true score difference $\theta^{\star\top}(x^\star - w_t)$. Let's now understand our regret definition for the three settings (Sec 2.1,2.2 and 2.3):
> >
> > - *For the dueling/pairwise preference (Logit-DB) feedback setup (Sec 2.1)*: Since the learner plays two decisions $(x_t,y_t)$ at each round $t$, the averaged (across the score of two decisions $(x_t,y_t)$) penalty of the learner becomes: $\theta^{\star\top}(x^\star - (x_t+y_t)/2)$. Thus, summing over $t = 1,\ldots,T$, the total regret across $T$ rounds becomes:
> > $$
> > R_T^{Logit-DB} = \sum_{t = 1}^T [\theta^{\star\top}(x^\star - (x_t+y_t)/2)],
> > $$
> > which is the same as our first regret objective as defined in Sec 2.1 (see Rem1).
> >
> > - *For the $B$-batched pairwise preference (Batched-Logit-DB) feedback setup (Sec 2.2):* In this case, following exactly *"the same penalty across averaged score-convention"* as adapted for the above regret, note the averaged (across the score of $2B$ decision points {$(x_t^i,y_t^i)$} (for $i \in [B]$) penalty of the learner becomes: $\theta^{\star\top}(x^\star - \frac{\sum_{i = i}^{B}(x_t^i+y_t^i)}{2B})$. Thus, summing over $t = 1,\ldots,T$, the total regret across $T$ rounds becomes:
> > $$
> > R_T^{B-LogitDB} = \sum_{t = 1}^T \Big[\theta^{\star\top}(x^\star - \frac{\sum_{i = i}^{B}(x_t^i+y_t^i)}{2B})\Big],
> > $$
> > which is the same as our batched regret objective as defined in Sec 2.2 (see Eq3).
> >
> > - Finally *For the MNL Bandits (top-$m$ ranking feedback $\sigma_{m,t}$) feedback setup (Sec 2.3):* Here again, following exactly *"the same penalty across averaged score-convention"*, note the averaged (across the $k$ decision points $\{(x_t^1,\ldots,x_t^k)\}$) penalty of the learner becomes: $\theta^{\star\top}(x^\star - \frac{\sum_{i = i}^{k} x_t^i}{k})$. And similarly as above, now summing over $t = 1,\ldots,T$, the total regret across $T$ rounds becomes:
> > $$
> > R_T^{MNL} = \sum_{t = 1}^T \Big[\theta^{\star\top}(x^\star - \frac{\sum_{i = i}^{k} x_t^i}{k})\Big],
> > $$
> > which is the same as our MNL regret objective as defined in Sec 2.3 (see Eq4).
> >
> >
> > Hopefully, that clarifies your concerns about the unification of the three regret bounds studied across the three feedback models, which should also justify the fairness of our performance measures. But please let us know if you have any follow-up questions. ``(Please note we are unable to make the symbols bold and calligraphic due to the poor latex interface)``
> >
> > > W3 (Q2): The paper claims computational efficiency compared to UCB or TS-based methods but does not provide evidence. Can the authors add numerical validation to support this claim?
> >
> > - Thanks for the great suggestion. We ran some experiments as part of our rebuttal, please see the common response. We added experimental evaluation in the updated draft (in the Openreview portal) and reported the regret performance with different types of feedback models studied in the paper, as well as the runtime performances.
> >
> > > Q1: In line 347, you claim your runtime requirement is \(O(dT)\). Does this account for the computational complexity of eigenvalue decomposition?
> >
> > - This is a great question and we should have added a discussion on  this. Please note, the computational complexity of eigendecomposition is only $O(d^3)$ which is just $\text{poly}(d)$ and efficient for any reasonable dimension $d$. To verify this, we have added runtime evaluation of our method in the updated draft. If you, please check the two tables in Sec 6.2 in the current Openreview draft, note we found that (on average) the algorithm takes only ~11.6 and ~17.4 secs to execute the algorithm for 50,000 and 80,000 rounds, respectively, for dimension ($d$) as large as $d=50$. Since each round requires computing exactly 1 eigendecomposition, this implies on average, the eigendecomposition for $d=50$ takes less than $0.0002475$ sec for $d=50$, which is minuscule. And *we ran all the experiments in a personal MacOS without using any computing servers/tools*. Clearly, this is the reason we are able to run our experiments for such large $d$ and $T$. Will add this explanation in the update.

---

> > > ### Author Response · Authors · 2024-12-02
> > > **Cont: Rebuttal for Reviewer rg2J**
> > >
> > > > Typos: Many thanks for pointing us to the typos.
> > >
> > > - Eq1: Yes, thanks so much. Please note we have corrected it in the updated draft.
> > >
> > > - Thm3 superscript: Yes, thanks. We have corrected it in the updated draft.
> > >
> > > - Line458: We meant “This property will play a very crucial role in our analysis, as we will see in the convergence proof of Thm5.” we have corrected it in the update.
> > >
> > > Thanks again for your careful reading and the great suggestions.
> > >
> > > ---
> > >
> > > Please let us know if you have any remaining questions, would be happy to clarify. We would greatly appreciate it if you could reconsider the scores based on the above clarifications.

---

> > > > ### Comment · Reviewer_rg2J · 2024-12-03
> > > >
> > > > I appreciate the author very much for the last-day response. I'm not sure whether the author fully understood my questions. I want to clarify that I don't have any concerns about this paper's connection to RLHF, nor did I put it as a concern in my initial review. I'm not sure why the authors had the opposite impression and wanted to justify the connection to RLHF (in my initial review Strength #1, I even wrote "the problem is of importance and directly related to solving RLHF problems").
> > > >
> > > > My main question is: why do we need to study adversarial human feedback in RLHF applications? Indeed it is more general than stochastic human feedback, but why do we need to leverage this generality? It is possible that algorithms for general settings do not work as well in special cases such as stochastic problems, so I do feel this question should be carefully discussed in the paper. The authors mentioned noise, fatigue, inattentiveness, and shift of preferences, but wouldn't there be better models for these problems, such as rotting (for fatigue and inattentiveness) or non-stationary stochastic (for shift of preference) settings? I feel the adversarial environment seems too pessimistic in terms of problem modeling. Some references from social science experiments may be helpful to justify this need.
> > > >
> > > > The comparison among settings is a minor presentation issue. It seems for $T$ rounds, different settings are getting different numbers of samples, i.e., $2T$ for logit-DB and $2BT$ for batched logit-DB. It seems to me that directly comparing regret is not so fair, makes more sense if the regret is compared at the level that all settings receive the same number of samples.
> > > >
> > > > Based on the response, I will keep my score (which is already positive).

---

> ### Author Response · Authors · 2024-12-03
> **Thank you**
>
> Dear Reviewer rg2J,
>
> We deeply appreciate your careful reading and maintaining a positive score for our paper. Thanks a lot.
>
> **Re. Motivation to RLHF** Sincere apologies for the misunderstanding. We interpreted your question about the "adversarial RLHF motivation" as addressing both the "adversarial model" and the "connection to RLHF and our problem." Thank you for the clarification.
>
> **Re. Adversarial Models** As you correctly pointed out, we rightfully wanted to capture the dynamic nature of human preferences through our adversarial model, but there are other alternative (and perhaps better for certain specific applications) ways to model the dynamicity, e.g. Robust (Corruptiontion) bandits, Rotting bandits, Sleeping bandits, Rising bandits, Dynamic regret, Contextual Bandits, etc. We absolutely agree with you that studying such a setting in the RLHF framework would be absolutely intriguing and beneficial for society.
>
> However, drawing motivation from the adversarial bandits and online learning literature (Auer et al,2002; Zinkevich,2003; Introduction to Online Convex Optimization by Hazan; and Bandit Algorithms by Lattimore & Szepesvári), we modeled the dynamic / shifting nature of the preference models through adversarial preferences, as no prior work addressed the framework in this setting. This could be beneficial for applications like e-commerce, ad-placement, online networks, and even recommender systems or language models, where it might be costly to deploy a personalized model for every user (as motivated by dynamic regret, sleeping, rotting or contextual bandits) but the platform is okay as to converge to the single best model that "overall serves all the T customers/rounds well on average". This is just one way to handle to shifting nature of real-world scenarios and by no way exhaustive. Certainly, extending to other more personalized scenarios are interesting open questions and lies in the wishlist of our future investigation.
>
> **Re. Regret expression for different settings:** Thanks a lot for clarifying your question further. This is a great question and we agree that indeed we are not treating all the rounds equal. So in our convention the B rounds of the batched feedback (Sec 2.2) is considered as 1 "round (sample) of batched feedback"; similarly, each top-$m$ ranking feedback $\sigma_{m,t}$ over $k$ items in $S_t$ (Sec 2.3) is considered to be 1 round (sample) of top-$m$ feedback. Hence, we penalize the averaged score in those $B$ or $m$ extracted samples.
>
> Now, if we wish to count the query time of $B$-batched feedback as $B$ rounds or the top-$m$ ranking query as $m$, indeed, the improved guarantees will no longer be valid. But this is also justified as all three settings become exactly the same; e.g., as you also correctly pointed out, the $B$-batched regret would simply become a Logit-DB problem on $BT$ rounds without any change in the regret measure convention.
>
> Our main point was to understand that if the learning algorithm has access to *"richer feedback"* per round (like $B$-batched feedback or top-$m$ rank-ordered feedback) *and without paying any extra cost per round for those richer feedback* (i.e. without changing the query cost per round), could the algorithms exploit this additional information and by what factor? As a result, we showed the improved performance guarantees with Thm 4,6,7,9 or Cor 5,8. Indeed the assumption of richer feedback but at the same per round cost is crucial behind those improved bounds. We will certainly clarify this in the updated version of the draft. Thanks for probing this further and we hope that clarifies your concern. If you have any other interesting observations in this regard or have another idea to formulate the performance measures for these richer feedback models, please let us know. That would be interesting to think through.
>
> Thanks again for all the feedback and suggestions.
>
> Sincerely,
> The authors

---

### Official Review · Reviewer_ETtp · 2024-11-01

**Soundness:** 2
**Presentation:** 3
**Contribution:** 2
**Rating:** 6
**Confidence:** 3

**Summary:**

This paper introduces Double-Scrible, a gradient descent algorithm for online linear optimization with adversarial preferences applicable to the online alignment of LLMs. Using mirror descent with a self-concordant barrier function, it achieves near-optimal regret bounds and extends to batched and partial ranking feedback.

**Strengths:**

The paper is well written. Both the problem and algorithms are rigorously explained, and the notation is intuitive.  I did not check the proofs, but the claims in the paper are sound. I appreciate various practical extensions of the algorithm to batched and ranked settings.

**Weaknesses:**

The algorithm is not empirically evaluated. Without any experiments, it may be difficult for the readers to apply the algorithm in practice as there is no reference implementation. Furthermore, aligning LLMs is a difficult domain, and theoretically sound algorithms may not necessarily have an effect as the human preferences are non-linear and context-dependent, the number of LLM arms grows exponentially with the length of the sentence, etc. Even small-scale experiments using, e.g., GPT-2, can increase the impact of the paper.

**Questions:**

Minor comment: Undefined reference at L458.

---

> ### Author Response · Authors · 2024-12-02
> **Rebuttal for Reviewer ETtp**
>
> Thanks for your valuable feedback. We clarify your specific questions below:
>
> > W1: Without empirical evaluation, it is difficult to assess the practicality of the proposed algorithm. Can the authors include experiments, even small-scale ones (e.g., with GPT-2), to increase the paper's impact?
>
> - Thanks for the great suggestion. Please refer to our common response. We added experimental evaluation in the updated draft (in the Openreview portal) and reported the regret performance with different types of feedback models studied in the paper, as well as the runtime performances.
>
> > Q1: Minor comment: Undefined reference at Line 458.
>
> - Sorry for the typo, we wanted to write, “This property will play a very crucial role in our analysis, as we will see in
> the convergence proof of Thm5.” Thanks for pointing, will correct it in the update.
>
> ---
>
> Please let us know if you have any remaining questions, would be happy to clarify. We would greatly appreciate it if you could reconsider the scores based on the above clarifications.

---

> > ### Comment · Reviewer_ETtp · 2024-12-02
> >
> > I want to thank the authors for their hard work during the rebuttal and adding the experiments. The new synthetic experiments are valuable as they empirically prove the $\mathcal{O}(d\sqrt{T})$ regret. Because of that, I am raising my score from 5 to 6. However, I still believe that the authors should evaluate their method and other baselines on proper RLHF experiments in the next version of the paper, as this is the primary motivation used throughout the paper. If I understand this correctly, the new synthetic experiments are designed in a way (continuous decision space) that related work cannot be applied. This, however, would not be the case in RLHF (generating text is not a continuous problem). I understand this is not feasible in such a short time during the rebuttal period.

---

> > > ### Author Response · Authors · 2024-12-02
> > > **Thank you**
> > >
> > > Dear Reviewer ETtp,
> > >
> > > Thank you so much for your consideration; we really appreciate your understanding and also the additional feedback. Indeed, conducting real-world LLM experiments with open-sourced LLM datasets and RLHF baselines lies beyond the scope of the present work, given the limited time and nature of this work; but we certainly want to extend our results and theoretical guarantees to RL setting as part of our future studies along with the real-world experiments as you suggested.
> > >
> > > Thanks a lot for your consideration and suggestions again. Sincerely,
> > > Authors

---

### Official Review · Reviewer_iPuV · 2024-11-03

**Soundness:** 3
**Presentation:** 3
**Contribution:** 2
**Rating:** 6
**Confidence:** 3

**Summary:**

This paper proposes an efficient online mirror descent algorithm called "Double-Scrible" for optimizing Large Language Models (LLMs) using human preference feedback, where the algorithm only observes relative preferences between pairs of outputs rather than direct feedback.

**Strengths:**

1. This paper seems to be the first to address the problem of adversarial online linear optimization with preference feedback, and the analysis is very comprehensive. Some techniques such as the gradient estimation for preference feedback seem novel to me.
2. The algorithms/assumptions and Theorems are clearly stated.

**Weaknesses:**

1. Missing related work section? And adding some intuition on algorithm design would make it easier for the readers to understand.
2. The paper assumes linear model, and the algorithms require eigenvalue decomposition, which makes the algorithms hard to scale up.

**Questions:**

1. The paper assumes linear model. Is it possible to extend it to the nonlinear setting? And is there a way to circumvent the eigenvalue decomposition?
2. Since the proposed algorithms are online, I wonder how the algorithms facilitate exploration.

**Details Of Ethics Concerns:**

It's a theoretical paper and don't have ethics concerns.

---

> ### Author Response · Authors · 2024-12-02
> **Rebuttal for Reviewer iPuV**
>
> Thanks for your valuable feedback.  We clarify your questions below:
>
> > W1: Related work section and additional intuition on algorithm design.
>
> **(I) Re. Related works:** Thanks for the suggestion. We included a discussion of the related work in the section titled "Limitations of Existing Online RLHF with Preference Feedback Algorithms''.  To make it clear we would be happy to rename that section to "Related Works'' and also include any additional reference that you might have in mind.
>
> **(II) Intuitive algorithm explanation:**  Thanks again for the suggestion. Our first and main algorithm, Double-Scrible (Alg 1) can be seen as a variant of the Follow-the-Regularized-Leader (FTRL) framework (https://haipeng-luo.net/courses/CSCI699/lecture2.pdf) that aims to predict the action with least cumulative regularized loss. Note in our case, this boils down to the prediction step:
>
> $$
> w_t= \arg\min_{w \in D_{\delta}} [ \eta \sum_{\tau =1 }^{t-1} (-g_\tau)^\top w +  \psi(w)  ],
> $$
> where $g_t$ denotes the estimated parameter $\theta_t^\star \in R^d$ as proved in Lem 11: Hence given any fixed $w \in D$, the first term in the above optimization, i.e. $\sum_{\tau =1 }^{t-1} (-g_\tau)^\top w$, simply denotes the total cumulative loss of the decision point $w \in \mathcal{D}$, while the second term with the $\nu$-self concordant function $\psi(w)$ acts as the regularization making the entire objective an FTRL optimization. ``(Note we're unable to make the symbols bold and calligraphic due to the poor latex interface)``
>
> It is also worth noting that FTRL algorithms can be seen as an online convex optimization (OCO) method, and an online variant of the well known classical Mirror Descent (OMD) algorithm (Nemirovsky \& Yudin, 1983), as observed by Kale \& Hazan'08 (https://www.satyenkale.com/papers/var-journal.pdf). Note the algorithms of Online Mirror Descent (OMD) are well researched in the online learning and online convex optimization literature [Intro to Online Optimization, Elad Hazan (2016), Online Convex Optimization, Shai Shalev-Shwartz (2012)]. The OMD algorithms are essentially a generalization of online gradient descent (OGD) algorithms [Zinkevich'03], where note our above optimization problem exactly boils down to an online gradient descent step for the choice of $\psi(\mathbf{w}) = \|\mathbf{w}\|_2^2$.
>
> Unlike UCB or TS approaches, OGD (or more generally OMD) algorithms are not based on *explore-exploit* ideas. The explore-exploit tradeoff arises when an algorithm must balance between (i) Exploration: Gathering information to improve future decisions and (ii) Exploitation: Leveraging current knowledge to make the best decision based on past information which however does not directly aim to minimize a convex loss or utilize the (full/estimated) gradient information if available. On the other hand, OGD (or generally OMD) is a deterministic algorithm for online convex optimization, where the goal is to minimize a sequence of convex losses over time (i) At each step, the algorithm updates its decision using the gradient of the loss function at the current point, and (ii) It incorporates regularization (via projections or constraints) to ensure decisions remain feasible. These two steps combined yield the FTRL step we suggested above. These algorithms are especially ideal for adversarial settings, as they aim to minimize regret by following the steepest descent direction iteratively and achieve optimal regret guarantees (Kale \& Hazan'08).
>
> | **Aspect**               | **Explore-Exploit**                     | **OMD**           |
> |---------------------------|-----------------------------------------|---------------------------------------------|
> | Feedback              | Partial (e.g., reward for one action)  | (full or estimated) gradient or loss function   |
> | Primary Goal          | Balance exploration and exploitation   | Minimize regret over convex losses         |
> | Stochasticity        | Often probabilistic or randomized      | Deterministic updates                      |
> | Usage Context         | OL, Bandits (esp. for finite actions)        | Online convex optimization (for cont. decision spaces)                 |
> | Focus                 | Reducing uncertainty over time         | Converging to the optimal solution with gradient descent |
>
> Hope that addressed your concerns about the intuition behind our algorithm as well as the *exploration step*.
>
> Our other algorithm, *Bable-Scrible* (Alg2), is essentially a variant of our Alg1 (*Double-Scrible*) that ensures an *improved (low-variance) $\boldsymbol{\theta}_t^\star$* estimate as shown in the aggregated $g_t$ estimate in Line 11, Alg2 using $B$-batched feedback. This is also justified in Lem 12 & Lem 13. Our third algorithm, MNL-Scrible (Alg3), essentially exploits the batched-estimate technique of Alg2 after extracting $O(m)$-pairwise feedback from the top-$m$ ranking feedback $\sigma_{m,t}$. We already provided a detailed description of Alg3 in Sec 5.

---

> ### Author Response · Authors · 2024-12-02
> **Contd: Rebuttal for Reviewer iPuV**
>
> > W2: The eigenvalue decomposition requirement makes the algorithm hard to scale. Are there ways to address this scalability issue?
>
> - This is a great question and we should have added a discussion on  this. Please note, the computational complexity of eigendecomposition is only $O(d^3)$ which is $\text{poly}(d)$ and efficient for any reasonable dimension $d$. We added this explanation in the updated draft. Please note we have also reported the experiments in the updated version of the draft (in the Openreview portal) with runtime guarantees that show how fast they run (order of just a few seconds for $d$ as large as $50$!) on standard Mac OS. We also reported regret performance with different types of feedback models studied in the paper, as well as the runtime performances.
>
> > Q1: The paper assumes a linear model. Is it possible to extend it to the nonlinear setting?
>
> - This is a great question and certainly an exciting direction to work in the future but beyond the scope of the current work. If the scoring function is linear, the overall idea could be to linearize convex functions as;
> $$
> f(x) - f(y) \leq \langle\nabla f(x),(x-y)\rangle.
> $$
> However compared to the linear functions the analysis would be harder since estimating $\nabla f(x)$ becomes harder and consequently the regret analysis becomes more subtle. For any general non-convex functions, optimization is a more challenging task because they may have multiple local minima and maxima, and not all optimization algorithms guarantee convergence to the global minimum. But one could potentially try to blend our OMD approach with the existing non-convex optimization techniques towards local convergence. But these are very broad potential areas of research and would definitely be interesting investigate in future.
>
> > Q2: Since the proposed algorithms are online, how do they facilitate exploration?
>
> - That is also a great question. Exploration is taken care of with the online mirror descent (OMD) logic which adapts to a regularized follow the regularized leader (FTRL) algorithmic logic. We detailed this in our *response to W1* above, kindly go over that. In summary the OMD algorithm proposed in our work (Alg 1,2,3) are not exploration-exploitation based (unlike UCB or TS), but gradient descent based, which is a classical method for optimizing convex functions. OMD methods aim to improve the current estimate $(w_t)$ of the optimum decision point $(x^\star)$ by making small steps towards the direction of the gradient (for maximization problems) which is the variant of the exploration step in OGD (or more generally OMD) types algorithms.
>
> ---
>
> Please let us know if you have any remaining questions, would be happy to clarify. We would greatly appreciate it if you could reconsider the scores based on the above clarifications.

---

> > ### Comment · Reviewer_iPuV · 2024-12-02
> >
> > Thank you for your response. I have no further questions and will maintain my (positive) score.

---

> > > ### Author Response · Authors · 2024-12-03
> > > **Thank you**
> > >
> > > Dear Reviewer iPuV,
> > >
> > > Thanks for the feedback and maintaining your positive evaluation of our work. We would, of course, be happy to address any follow-up questions or clarifications in the remaining time, should anything come to your mind.
> > >
> > > Sincerely,
> > > Authors

---

### Official Review · Reviewer_gxKZ · 2024-11-05

**Soundness:** 3
**Presentation:** 3
**Contribution:** 2
**Rating:** 5
**Confidence:** 2

**Summary:**

The paper "Dueling in the Dark: An Efficient and Optimal Mirror Descent Approach for Online Convex Optimization with Adversarial Preferences" addresses the challenge of using human preference feedback in reinforcement learning, particularly with applications for AI alignment in large language models (LLMs). The main focus is on creating an efficient online gradient descent algorithm capable of handling adversarially changing preferences, providing theoretical guarantees for performance.

**Strengths:**

1. Algorithmic Innovation: The authors introduce an Online Mirror Descent (OMD)-based algorithm, named Double-Scrible, that optimally handles adversarial preferences using only weak preference feedback, unlike previous models that rely on value feedback.

2. Performance Guarantees: This algorithm achieves optimal regret bounds and includes generalizations to batched feedback (handling multiple preference queries at once) and multi-way preference feedback (handling partial ranking).

3. Efficiency: The approach offers computational efficiency, particularly suitable for high-dimensional and adversarial environments, making it more practical for real-world AI applications.

**Weaknesses:**

1. For the formulation, there is already some work formulating the problem like a dueling bandit (Xiong et al. (2024), [2]) and even dueling bandit with human feedback [1]. I think this work needs to provide more discussion of the comparison with those works. Especially with [1], because they both consider adversarial preference feedback.

2. One of the contributions that the authors highlight is the computational efficiency of their algorithm, but they don't carry out experiments or even simulations to show the computational efficiency of their algorithm. It is questioned whether their algorithm can be implemented. Hence, the authors are suggested to provide some experiments.


[1] Di Q, He J, Gu Q. Nearly optimal algorithms for contextual dueling bandits from adversarial feedback[J]. arXiv preprint arXiv:2404.10776, 2024.

[2] Wang Y, Liu Q, Jin C. Is RLHF More Difficult than Standard RL?[J]. arXiv preprint arXiv:2306.14111, 2023.

**Questions:**

1. Line 458: "in the convergence proof of ??" lacks the reference.

2. Although the bound for top m ranking is better than pairwise comparison, the query times of the two methods are different. Can the authors compare the total query complexity of the two methods to achieve the same suboptimality?

---

> ### Author Response · Authors · 2024-12-01
> **Rebuttal for Reviewer gxKZ**
>
> Thanks for your valuable feedback. We clarify your specific questions below:
>
> > W1: Comparison with related works:
>
> - [1] suffers from the same limitation mentioned in Line 082-084 -- the algorithm of [1] is based on the same optimism in the face of the uncertainty (UCB) principle that requires maintaining confidence sets and optimizing over the policy space, which is computationally intractable. Specifically, Line 6 of their Alg 1 is computationally intractable for continuous decision spaces. The paper also does not explain how to find the dueling pair $(a_t,b_t)$ in Line 6 of Alg1.
>
> - [2] The objective of [2] is  to find an $\epsilon$-optimal policy (see Sec 2.1 of [2]), which is way more simple than the dueling regret objective considered in our paper. This is because, in the objective of [2], there is no penalty for playing both arms; it is a pure exploration setting and reduces the problem to a reward-based generalized linear (GLM) bandit  (a.k.a.\ logistic bandit) problem. In other words, [2] allows the algorithm to play arbitrarily bad arms without any penalty as long as they are able to find the best arm (policy) with a `small' number of observations, while our regret objective requires a balanced exploration-exploitation, which requires us to find the best-policy while being optimal in both arms -- hence our objective is more difficult than that considered in [2]. Consequently, our problem can not be directly reduced to a standard bandit framework, unlike that used in [2], and hence, our approach requires significant novelty in terms of the algorithm design as well as in the proof analysis.
>
> Nevertheless, we appreciate the pointers and will be happy to cite them in the final version.
>
> > W2: The authors claim computational efficiency but do not provide experiments or simulations to demonstrate this. Can the authors validate their computational efficiency claims with experimental evidence?
>
> - Please see the common response. We added experimental evaluation in the updated draft (in the Openreview portal) and reported the regret performance with different types of feedback models studied in the paper, as well as the runtime performances. If you, please check the two tables in Sec 6.2 in the current Openreview draft, note we found that (on average) the algorithm takes only ~11.6 and ~17.4 secs to execute the algorithm for 50,000 and 80,000 rounds, respectively, for dimension ($d$) as large as $d=50$. Since each round requires computing exactly 1 eigendecomposition, this implies on average, the eigendecomposition for $d=50$ takes less than $0.0002475$ sec for $d=50$, which is minuscule. And *we ran all the experiments in a personal MacOS without using any computing servers/tools*. Clearly, this is the reason we are able to run our experiments for such large $d$ and $T$. Will add this explanation in the update.
>
> > Q1: Line 458: "in the convergence proof of ??" lacks the reference.
>
> - Sorry for the typo, we wanted to write “This property will play a very crucial role in our analysis, as we will see in
> the convergence proof of Thm5.” Thanks for pointing, will correct it in the update.
>
> > Q2: Although the bound for top-m ranking is better than pairwise comparison, the query times of the two methods are different. Can the authors compare the total query complexity of the two methods to achieve the same suboptimality?
>
> - This is a great question. please note by our convention, each ranking top-m query is counted as one query. As described in Lines 491-493 -- our algorithm MNL-Scrible manages to extract $m$ independent pairwise preference feedback from $\sigma_{m,t}$, which implies one top-$m$ ranking query can represent $m$ independent pairwise preference queries. So if the goal is to find the best-arm (or $\epsilon$-best arm) the sample complexity of the problem reduces by a factor of $m$ compared to achieving that with just pairwise preferences.
>
> Now if we wish to count the query time of top-$m$ ranking query as $m$ times that of the dueling query, indeed the query cost of both feedback model becomes the same (for a given suboptimality $\epsilon>0$). Hope that clarifies your questions.
>
> ---
>
> Please let us know if you have any remaining questions, would be happy to clarify. We would greatly appreciate it if you could reconsider the scores based on the above clarifications.

---

> ### Author Response · Authors · 2024-12-03
> **Follow-up**
>
> Dear Reviewer gxKZ,
>
> Thank you once again for taking the time to review our paper and provide thoughtful and valuable feedback. We hope that our rebuttal has addressed your concerns.
>
> As the discussion period approaches its conclusion, we kindly ask if we could address any additional questions or comments that we could clarify in the remaining time.
>
> If you think that our responses have sufficiently addressed your concerns, we would be truly grateful if you could reconsider the evaluation of our paper based on the clarification provided in the rebuttal.
>
> Thanks, Authors

---

### Author Response · Authors · 2024-12-01
**Common Response: Experimental Evaluation added in the updated draft**

We wanted to thank all the reviewers for their careful reading and insightful comments. As suggested by most of the reviewers, we have conducted detailed experiments to evaluate the empirical performances of our proposed methods.

We added the detailed discussion of the empirical performance of our algorithm in Sec 6, along with its regret and runtime guarantees with varying $d, B$ and $m$.

Please go over the plots in the updated draft to check the empirical results. For completeness, we further add the *details of the experiments below*:

We run synthetic experiments to report the performance of our methods, Double-Scrible (Sec 3) and MNL-Scrible (Sec 5) respectively, for dueling and top-$m$ ranking feedback. All results are averaged across $100$ runs. {It is important to note that no existing algorithms that address the problem of Logit-DB efficiently apply to continuous decision spaces. Existing approaches suggested by Di et al'2024, Bengs et al' 2022 get computationally infeasible for large or continuous action space due to which they reported experiments only on finite decision spaces. Also note that some of the prior works, e.g. Kausik at al'2024, Saha et al'2023, do not report any experiments due to the computational inefficiency of their algorithms. We are hence unable to compare our results, which are reported on continuous decision spaces, with any prior results due to their computational inefficiency. We run our experiments on different environments (adversarial loss sequences):

**Adversarial $\{\boldsymbol \theta}_t^\star$ Environments.**
We report our experiment results on problem instances with varying dimension $d$ generated as follows:
(1) **Inst**-$1(d)$: For a given dimension $d$ and round $t$, we choose the adversarial sequence of score functions $\boldsymbol \theta_t^* \in {\mathbb R}^d$ such that for any co-ordinate $i \in [d]$, we first select
$$
\tilde \theta_t(i) = \begin{cases}
            \text{Unif}(0,1) - 0.37 ~\text{ if } i \leq \lceil{d/2}\rceil
            \\
            \text{Unif}(0,1) + 0.53 \text{ otherwise, }
            \end{cases}
$$
and set $\boldsymbol \theta_t^* = \boldsymbol{\tilde \theta_t}/{\|\boldsymbol{\tilde \theta_t}\|}$.

(2) **Inst**-$2(d)$: For this instance, again given a fixed $d$ and round $t$, we first choose $\boldsymbol{\tilde \theta}_t$ as:
$$
\tilde \theta_t(i) =
            \text{Unif}(0,1) + 0.57X ~\text{ for all } i \in [d],
$$
where $X \sim 2 \text{Ber}(0.5) -1 $ represents a $\{\pm 1\}$-valued binary random variable, and further and set $\boldsymbol \theta_t^* = \boldsymbol{\tilde \theta_t}/{\|\boldsymbol{\tilde \theta_t}\|}$.

The decision space is given by ${\mathcal D} = {{ \mathbf x \mid \mathbf{A} \mathbf{x} \leq \mathbf{b}} }$, for some ${\mathbf A} \in {\mathbb R}^{c \times d}$ and ${\mathbf b}\in {\mathbb R}^c$, for some $c \in {\mathbb N}_+$. For our purpose we select **A** s.t. **A** = [$I_d$; - $I_d$ ; -$(1_d)^\top$; $(1_d)$^\top]$ and **b**= $[0.951_d; 1_d; 0.45d; 0.45d]$.

**Choice of the self concordant barrier $\psi$:** We use the following self concordant barrier
$
\psi({\mathbf w}) = -\sum_{j=1}^c \ln(b_j - {\mathbf a}_j^\top {\mathbf w}), \text{ for any } {\mathbf w} \in {\mathbb R}^d,
$
which is known to be an $c$-self-concordant barrier for ${\mathcal D}$, ${\mathbf a}_j$ being the $j$-th row of ${\mathbf A}$ and $b_j$ denotes the $j$-th co-ordinate of ${\mathbf b}$ [Haipeng Luo, Lec 17].

---

### Meta-Review · Area_Chair_sSM8 · 2024-12-17

**Metareview:**

The submitted paper is high quality. The reviewers agree on this central premise but there seems to be disagreement on the applicability of the results for the domains scoped, as well as whether the theoretical results make sense in those contexts. On top of that, the paper seems to be violating some formatting requirements in terms of length restriction (core technical content is spilling onto the 11th page). For that reason, this paper needs to be further revised before it is suitable for publication.

**Additional Comments On Reviewer Discussion:**

See above.

---

### Decision · Program_Chairs · 2025-01-22

Reject